# Edge-Colored Clustering in Hypergraphs: Beyond Minimizing Unsatisfied Edges

Alex Crane [1]   Thomas Stanley [2]   Blair D. Sullivan [1]   Nate Veldt [2]

## Abstract

We consider a framework for clustering edge-colored hypergraphs, where the goal is to cluster (equivalently, to *color*) objects based on the primary type of multiway interactions they participate in. One well-studied objective is to color nodes to minimize the number of *unsatisfied* hyperedges – those containing one or more nodes whose color does not match the hyperedge color. We motivate and present advances for several directions that extend beyond this minimization problem. We first provide new algorithms for maximizing *satisfied* edges, which is the same at optimality but is much more challenging to approximate, with all prior work restricted to graphs. We develop the first approximation algorithm for hypergraphs, and then refine it to improve the best-known approximation factor for graphs. We then introduce new objective functions that incorporate notions of balance and fairness, and provide new hardness results, approximations, and fixed-parameter tractability results.

## 1. Introduction

Edge-colored clustering (ECC) is an optimization framework for clustering datasets characterized by categorical relationships among data points. The problem is formally encoded as an edge-colored hypergraph (Figure 1), where each edge represents an interaction between data objects (the nodes) and the color of the edge indicates the *type* or *category* of that interaction. The goal is to assign colors to nodes in such a way that edges of a color tend to include nodes of that color, by minimizing or maximizing some objective function relating edge colors and node colors. ECC algorithms have been applied to various clustering tasks where cluster labels naturally match with interaction types. For

[1]Kahlert School of Computing, University of Utah, Salt Lake City, USA [2]Department of Computer Science and Engineering, Texas A&M University, College Station, USA. Correspondence to: Alex Crane <alex.crane@utah.edu>.

*Proceedings of the 42nd International Conference on Machine Learning*, Vancouver, Canada. PMLR 267, 2025. Copyright 2025 by the author(s).

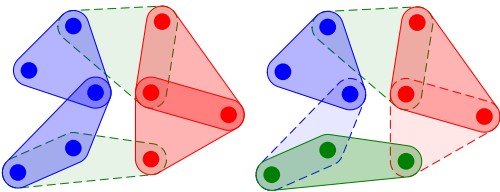

*Figure 1.* The node coloring on the left satisfies 4 edges (2 blue, 2 red, 0 green). The coloring on the right only satisfies 3, but each color has a satisfied edge.

example, if nodes are researchers, edges are author lists for publications, and colors indicate publication field (computer science, biology, etc.), then ECC provides a framework for inferring researchers' fields based on publications. ECC has also been used for temporal hypergraph clustering (Amburg et al., 2020), where edge colors encode time windows in which interactions occur. ECC then clusters nodes into time windows in which they are especially active. Variants of ECC have also been used for team formation (Amburg et al., 2022), in which case nodes are people, edges represent team tasks, and colors indicate task type. In this setting, ECC corresponds to assigning tasks based on prior team experiences.

**Related work and research gaps.** Edge-colored clustering has been well-studied in the machine learning and data mining literature from the perspective of approximation algorithms (Amburg et al., 2020; 2022; Veldt, 2023; Crane et al., 2024; Angel et al., 2016) and fixed-parameter tractability results (Kellerhals et al., 2023; Cai & Leung, 2018; Crane et al., 2024). It is also closely related to chromatic correlation clustering (Bonchi et al., 2012; 2015; Anava et al., 2015; Klodt et al., 2021; Xiu et al., 2022), which is an edge-colored variant of correlation clustering (Bansal et al., 2004). Several variants of ECC have been encoded using different combinatorial objective functions for assigning colors to nodes. The earliest and arguably the most natural is MAXECC (Angel et al., 2016), which seeks to maximize the number of *satisfied* edges—edges in which all the nodes match the color of the edge. More recent attention has been paid to MINECC (Amburg et al., 2020), where the goal is to minimize the number of *unsatisfied* edges. The latter is the same as MAXECC at optimality but differs in terms of approximations and fixed-parameter tractability results. The approximability of MINECC is currently well-understood: a recent ICML paper designed improved approximation guarantees that are tight with respect to linear

programming integrality gaps and nearly tight with respect to approximation hardness bounds (Veldt, 2023). However, existing approximations for MAXECC apply only to graph inputs, whereas nearly all algorithms for MINECC apply to hypergraphs. There is also a much larger gap between the previous best 0.3622-approximation algorithm (Ageev & Kononov, 2020) and the best existing upper bound, which only rules out approximation factors better than 0.972 assuming P $\neq$ NP (Alhamdan & Kononov, 2019).

Another limitation of prior work is that nearly all objective functions for ECC (including MINECC and MAX-ECC) only consider the total number of edges that are (un)satisfied, even if it means certain colors are disproportionately (un)satisfied. Figure 1 provides a small example where the optimal MINECC solution satisfies four edges but only from two colors. Meanwhile, another color assignment satisfies only three edges but includes a satisfied edge of each color. The latter choice is natural for applications where one may wish to incorporate some notion of balance or fairness in edge satisfaction. For example, if coloring nodes means assigning individuals to tasks for future team interactions, then we would like to avoid situations where one type of task is assigned no workers. Balanced and fair objectives have been studied and applied for many other recent clustering frameworks, but have yet to be explored in the context of edge-colored clustering. As a key example, there have been many recent advances in algorithms for alternative objectives for correlation clustering, which is closely related to ECC. This includes results for minimizing the maximum number of edge "mistakes" incident to any one node (Davies et al., 2023; Heidrich et al., 2024; Cao et al., 2025), as well as algorithms for a generalized $\ell_p$-norm objective (Puleo & Milenkovic, 2016; Kalhan et al., 2019; Jafarov et al., 2021; Charikar et al., 2017; Davies et al., 2024) that captures the latter mini-max objective when $p = \infty$ and captures standard correlation clustering when $p = 1$.

Motivated by the above, we present new contributions to ECC along two frontiers: improved MAXECC algorithms, and new frameworks for balanced and fair ECC variants.

**Our contributions for** MAXECC. We present the first approximation algorithm for hypergraph MAXECC, which has an approximation factor of $(2/e)^r (r + 1)^{-1}$ where $r$ is the maximum hyperedge size. The approximation factor goes to zero as $r$ increases, but this is expected since a prior hardness result rules out the possibility of constant-factor approximations that hold for arbitrarily large $r$ (Veldt, 2023). Our result shows that non-trivial constant-factor approximations can be obtained when $r$ is a constant. We use insights from our hypergraph algorithm to obtain a new best approximation factor of $154/405 \approx 0.38$ for the graph version of the problem, improving on the previous best 0.3622-approximation shown five years ago (Ageev & Kononov,

2020). While the increase in approximation factor appears small at face value, this improves on a long sequence of papers on MAXECC for the graph case (Angel et al., 2016; Ageev & Kononov, 2015; Alhamdan & Kononov, 2019; Ageev & Kononov, 2020). Obtaining even a minor increase in the approximation factor is highly non-trivial and constitutes the most technical result of our paper.

**Our contributions for balanced and fair ECC variants.** Our first contribution towards balanced and fair variants of ECC is to introduce the generalized $\ell_p$-NORM MINECC objective, which uses a parameter $p$ to control balance in unsatisfied edges, and captures MINECC as a special case when $p = 1$. For $p = \infty$, the objective corresponds to a new problem we call COLOR-FAIR MINECC, where the goal is to minimize the maximum number of edges of any color that are unsatisfied. We prove COLOR-FAIR MINECC is NP-hard even for $k = 2$ colors, even though standard MINECC is polynomial-time solvable for $k = 2$. We provide a 2-approximation for $\ell_p$-NORM MINECC for every $p \geq 1$ by rounding a convex relaxation, and give a $2^{1/p}$-approximation for every $p \in (0, 1)$. We prove that this factor 2 for $p \geq 1$ is tight in the sense that it matches an integrality gap for the $p = \infty$ case. We then study COLOR-FAIR MINECC from the perspective of parameterized complexity, proving it is fixed-parameter tractable (FPT)[1] in terms of the total number of unsatisfied edges, but is not FPT (under standard complexity assumptions) for many other parameters including feedback edge number, cutwidth, treewidth + maximum degree, and slim tree-cut width; this is in contrast to positive results for the standard ECC objective (Kellerhals et al., 2023). We also consider a variant of COLOR-FAIR MINECC that maximizes the minimum number of satisfied edges of any one color, proving it is NP-hard to approximate to within any multiplicative factor.

We also consider scenarios in which the "importance" of the edge colors is unequal. In particular, we introduce PROTECTED-COLOR MINECC, where we are given a special color $c_1$ which can be thought of as encoding a protected interaction type. The goal is to color nodes to minimize the number of unsatisfied edges subject to a strict upper bound on the number of unsatisfied edges of color $c_1$. We give bicriteria approximation algorithms based on rounding a linear programming relaxation, meaning that our algorithm is allowed to violate the upper bound for the protected color by a bounded amount, in order to find a solution that has a bounded number of unsatisfied edges relative to the optimal solution. Finally, although the contributions of our paper are primarily theoretical, we also implement our algorithms for COLOR-FAIR MINECC and

---

[1] A problem is FPT with respect to parameter $k$ if it is solvable in $f(k) \cdot n^{O(1)}$ time, where $n$ is the instance size and $f$ is any computable function. See Appendix A for a detailed introduction.

PROTECTED-COLOR MINECC on a suite of benchmark ECC datasets. Our empirical results show that (1) our algorithms outperform their theoretical guarantees in practice, and (2) our balanced and fair objectives still achieve good results with respect to the standard MINECC objective while better incorporating notions of balance and fairness.

## 2. Improved MAXECC Algorithms

**Preliminaries.** For a positive integer $n$, let $[n] = \{1, 2, \ldots n\}$. We use bold lowercase letters to denote vectors, and indicate the $i$th of entry of a vector $\mathbf{x} \in \mathbb{R}^n$ by $x_i$. For a set $S$ and a positive integer $t$, let $\binom{S}{t}$ denote all subsets of $S$ of size $t$. An instance of edge-colored clustering is given by a hypergraph $H = (V, E, \ell, \{\omega_e\}_{e \in E})$ where $V$ is a node set, $E$ is a set of hyperedges (usually just called *edges*), and $\ell \colon E \to [k]$ is a mapping from edges to a color set $[k] = \{1, 2, \ldots, k\}$. For $e \in E$, we let $\omega_e \geq 0$ denote a nonnegative weight associated with $e$, which equals 1 for all edges in the unweighted version of the problem. For a color $c \in [k]$, let $E_c \subseteq E$ denote the edges of color $c$. We use $r$ to denote the *rank* of $H$, i.e., the maximum hyperedge size.

The goal of ECC is to construct a map $\lambda \colon V \to [k]$ that associates each node with a color, in order to optimize some function on edge *satisfaction*. Edge $e \in E$ is *satisfied* if $\ell(e) = \lambda(v)$ for each $v \in e$, and is otherwise *unsatisfied*. The two most common objectives are maximizing the total weight of satisfied edges (MAXECC) or minimizing the total weight of unsatisfied edges (MINECC). Formally, MAXECC can be cast as a binary linear program (BLP):

$$
\begin{aligned}
\max \quad & \textstyle\sum_{e \in E} \omega_e z_e \\
\text{s.t.} \quad & \forall v \in V : && \textstyle\sum_{c=1}^{k} x_v^c = 1 \\
& \forall c \in [k], e \in E_c : && x_v^c \geq z_e \quad \forall v \in e \\
& x_v^c, z_e \in \{0, 1\} && \forall c \in [k], v \in V, e \in E.
\end{aligned}
\tag{1}
$$

Setting $x_u^c$ to 1 indicates that node $u$ is given color $c$ (i.e., $\lambda(u) = c$). We use $\mathbf{x}_u = \begin{bmatrix} x_u^1 & x_u^2 & \cdots & x_u^k \end{bmatrix}$ to denote the vector of variables for node $u$. For edge $e \in E$, the constraints are designed in such a way that $z_e = 1$ if and only if $e$ is satisfied. A binary LP for MINECC can be obtained by changing the objective function to $\min \sum_{e \in E} \omega_e(1 - z_e)$.

**Challenges in approximating MAXECC.** Although MINECC and MAXECC are equivalent at optimality, the latter is far more challenging to approximate. Due to an approximation-preserving reduction from INDEPENDENT SET, it is NP-hard to approximate MAXECC in hypergraphs of unbounded rank $r$ to within a factor $|E|^{1-\varepsilon}$ (Veldt, 2023; Zuckerman, 2006). There also are simple instances (e.g., a triangle with 3 colors) where a simple 2-approximation of Amburg et al. (2020) for MINECC (round variables of an LP relaxation to 0 if they are strictly below $1/2$, otherwise round to 1) fails to satisfy *any* edges. These challenges

do not rule out the possibility of approximating MAXECC when $r$ is constant. Indeed there are many approximations for graph MAXECC ($r = 2$) (Angel et al., 2016; Alhamdan & Kononov, 2019; Ageev & Kononov, 2015; 2020), but these require lengthy proofs, rely fundamentally on the assumption that the input is a graph, and do not easily extend even to the $r = 3$ case. Here we provide a generalized approach that gives the first approximation guarantees for hypergraph MAXECC, when $r$ is constant. Our approach also provides a simplified way to approximate graph MAXECC, and we design a refined algorithm that achieves a new best approximation factor for graph MAXECC, improving on the long line of previous algorithms for this case.

### 2.1. Technical preliminaries for LP rounding algorithms

The LP relaxation for MAXECC can be obtained by relaxing the binary constraints in Binary Linear Program (1) to linear constraints $0 \leq x_v^c \leq 1$ and $0 \leq z_e \leq 1$. Solving this LP gives a fractional node-color assignment $x_v^c \in [0, 1]$. The closer $x_v^c$ is to 1, the stronger this indicates node $v$ should be given color $c$. For the remainder of this section, assume $\{z_e, x_v^c \colon e \in E, v \in V, c \in [k]\}$ specifically refers to an optimal set of LP variables for the relaxation. Our task is to round these variables to assign one color to each node in a way that satisfies provable approximation guarantees.

Our algorithms (Algorithms 1 and 2) are randomized. Both use the same random process to identify colors that a node "wants". For each $c \in [k]$ we independently generate a uniform random *color threshold* $\alpha_c \in [0, 1]$. If $x_u^c > \alpha_c$, we say that *node $u$ wants color $c$*, or equivalently that color $c$ wants node $u$. Because a node may want more than one color, we use a random process to choose one color to assign, informed by the colors the node wants. Our approximation proofs rely on (often subtle) arguments about events that are independent from each other. We begin by presenting several useful observations that will aid in proving our results.

Our first observation is that if we can bound the expected cost of every edge in terms of the LP upper bound, it provides an overall expected approximation guarantee.

**Observation 1.** *Let $\mathcal{A}$ be a randomized ECC algorithm and $p \in [0, 1]$ be a fixed constant. If for each $e \in E$ we have $\mathbb{P}[e \text{ is satisfied by } \mathcal{A}] \geq p z_e$, then $\mathcal{A}$ is a $p$-approximation.*

Let $X_u^c$ denote the event that $u$ wants $c$, and $Z_e$ be the event that every node in edge $e \in E$ wants color $c = \ell(e)$.

**Observation 2.** *For each node $v \in V$, the events $\{X_v^c\}_{c \in [k]}$ are independent, and $\mathbb{P}[X_v^c] = \mathbb{P}[\alpha_c < x_v^c] = x_v^c \leq 1$.*

**Observation 3.** $\mathbb{P}[Z_e] = \mathbb{P}[\cap_{v \in e} X_v^c] = \min_{v \in e} x_v^c = z_e$.

For an edge $e$ and color $i \neq \ell(e)$, we frequently wish to quantify the possibility that some node in $e$ wants color $i$, as this opens up the possibility that $e$ will be unsatisfied

**Algorithm 1** Approximation alg. for hypergraph MAXECC

Obtain optimal variables $\{z_e; x_v^c\}$ for the LP relaxation
$\pi \leftarrow$ uniform random ordering of colors $[k]$
For $c \in [k]$, $\alpha_c \leftarrow$ uniform random threshold in $[0, 1]$
**for** $v \in V$ **do**
    $\mathcal{W} = \{c \in [k]: \alpha_c < x_v^c\}$
    **if** $|\mathcal{W}| > 0$ **then**
        $\lambda(v) \leftarrow \text{argmax}_{c \in \mathcal{W}}\, \pi(c)$
    **else**
        $\lambda(v) \leftarrow$ arbitrary color
    **end if**
**end for**

because some node $v \in e$ is given color $i$. Towards this goal, for each $e \in E$ and color $i \in [k]$ we identify one node in $e$ that has the highest likelihood of wanting $i$. Formally, we identify some *representative* node $v \in e$ satisfying $x_v^i \geq x_u^i$ for every $u \in e$ (breaking ties arbitrarily if multiple nodes satisfy this), and we define $\sigma_e(i) = v$. The definition of $\sigma_e(i)$ implies the following useful observation:

**Observation 4.** *Color $i$ does not want **any** nodes in $e \iff$ color $i$ does not want $v = \sigma_e(i)$.*

Variations of Observations 1-3 have often been used to prove guarantees for previous randomized ECC algorithms. However, Observation 4 is new and is key to our analysis.

### 2.2. Hypergraph MaxECC Algorithm

In addition to color thresholds $\{\alpha_c\}$, our hypergraph MAX-ECC algorithm (Algorithm 1) generates a uniform random permutation $\pi$ to define priorities for colors. A node is then assigned to the highest priority color it wants.

**Theorem 2.1.** *Algorithm 1 is a $\left(\frac{1}{r+1}\left(\frac{2}{e}\right)^r\right)$-approximation algorithm for MAXECC in hypergraphs with rank $r$.*

*Proof.* Fix an arbitrary edge $e \in E$ and let $c = \ell(e)$. Let $T_e$ denote the event that $e$ is satisfied. By Observation 1, it suffices to show $\mathbb{P}[T_e] \geq \frac{z_e}{r+1}\left(\frac{2}{e}\right)^r$.

Let $C = [k] \setminus \{c\}$. To partition the color set $C$, for each $v \in e$ we define $C_v = \{i \in C: \sigma_e(i) = v\}$. Recall that $\sigma_e(i)$ identifies a node $v \in e$ satisfying $x_v^i = \max_{u \in e} x_u^i$. Let $A_v$ denote the event that at most one color in $C_v$ wants one or more nodes in $e$. From Observation 4, $A_v$ is equivalent to the event that $v$ wants at most 1 color in $C_v$. Thus, the probability of $A_v$ is the probability that at most one of the events $\{X_v^i: i \in C_v\}$ happens. Observation 2 gives $\sum_{i=1}^{k} \mathbb{P}[X_v^i] = \sum_{i=1}^{k} x_v^i = 1$, allowing us to repurpose a supporting lemma of Angel et al. (2016) on graph MAXECC to see that $\mathbb{P}[A_v] \geq 2/e$ (Lemma B.1 in Appendix B).

Because color thresholds $\{\alpha_i\}$ are drawn independently for each color, and because color sets $\{C_v: v \in e\}$ are disjoint

from each other and from $c$, the events $\{A_v, X_v^c: v \in e\}$ are mutually independent. Thus, using Observation 3 gives

$$\mathbb{P}\left[\left(\bigcap_{v \in e} A_v\right) \cap Z_e\right] = \mathbb{P}[Z_e] \cdot \prod_{v \in e} \mathbb{P}[A_v] \geq z_e \cdot \left(\frac{2}{e}\right)^r.$$

If the joint event $J = (\cap_{v \in e} A_v) \cap Z_e$ holds, this means every node in $e$ wants color $c$, and at most $r$ distinct other colors (one for each node in $v \in e$ since there is one set $C_v$ for each $v \in e$) want one or more nodes in $e$. Conditioned on $J$, $e$ is satisfied if the color $c$ has a higher priority (determined by $\pi$) than the other $r$ colors, which happens with probability $1/(r+1)$. Thus,

$$\mathbb{P}[T_e] \geq \mathbb{P}[T_e \mid J]\,\mathbb{P}[J] \geq \frac{z_e}{r+1}\left(\frac{2}{e}\right)^r.$$

$\square$

**Comparison with prior graph MAXECC algorithms.** Algorithm 1 achieves a $4/(3e^2) \approx 0.18$ approximation factor for graph MAXECC ($r = 2$). This improves upon the first ever approximation factor of $1/e^2 \approx 0.135$ for graph MAXECC (Angel et al., 2016), and comes with a significantly simplified proof. There are two interrelated factors driving this simplified and improved guarantee. The first is our use of Observation 4. Angel et al. (2016) bound the probability of satisfying an edge $(u, v) \in E_c$ using a delicate argument about certain dependent events we can denote by $B_v$ (the event that $v$ wants at most 1 color from $C = [k] \setminus \{c\}$) and $B_u$ (defined analogously for $u$). The algorithm of Angel et al. (2016) requires proving that $\mathbb{P}[B_u \cap B_v] \geq \mathbb{P}[B_u]\mathbb{P}[B_v]$, even though $B_u$ and $B_v$ are dependent. The proof of this result (Proposition 1 in Angel et al. (2016)) is interesting but also lengthy. It fundamentally relies on the assumption that there are exactly two nodes in the edge, and involves several long algebraic expressions to describe probabilities for different events. In its current form the result does not apply when $r > 2$. Attempting to generalize it even for $r = 3$ leads to far more complicated algebraic expressions, and even then it is unclear if a similar result can be shown. Subsequent approximation algorithms for graph MAXECC also rely on complicated variants of this proposition (Ageev & Kononov, 2015; 2020; Alhamdan & Kononov, 2019). In contrast, Theorem 2.1 deals with mutually independent events $\{A_v: v \in e\}$, which allows us to immediately see that $\mathbb{P}[\cap_{v \in e} A_v] \geq \prod_{v \in e} \mathbb{P}[A_v]$ for arbitrary sized hyperedges. Observation 4 provides the key insight as to why it suffices to consider these events when bounding probabilities, leading to a far simpler analysis that extends easily to hypergraphs.

Our second key factor is the use of a global color ordering $\pi$. Angel et al. (2016) and other results for graph MAXECC apply a two-stage approach where Stage 1 identifies which

**Algorithm 2** 0.38-approximation alg. for graph MAXECC

Obtain optimal variables $\{z_e; x_v^c\}$ for the LP relaxation
$\pi \leftarrow$ uniform random ordering of colors $[k]$
For $c \in [k]$, $\alpha_c \leftarrow$ uniform random threshold in $[0, 1]$
**for** $v \in V$ **do**
$\quad S_v \leftarrow \{c \in [k] \mid x_v^c \geq 2/3\}; W_v \leftarrow [k] \setminus S_v$
$\quad W_v' = \{c \in W_v \mid \alpha_c < x_v^c\}$
$\quad$**if** $|W_v'| > 0$ **then**
$\quad\quad \lambda(v) \leftarrow \mathrm{argmax}_{i \in W_v'} \pi(i)$
$\quad$**else if** $\exists c$ s.t. $S_v = \{c\}$ **then**
$\quad\quad \lambda(v) \leftarrow c$
$\quad$**else**
$\quad\quad \lambda(v) \leftarrow$ arbitrary color
$\quad$**end if**
**end for**

---

nodes "want" which colors, and Stage 2 assigns nodes to colors *independently* for each node. To illustrate the difference, consider the probability of satisfying an edge $(u, v) \in E_c$ if we condition on $u$ and $v$ both wanting $c$ and each wanting at most one other color. The algorithm of Angel et al. (2016) has a $1/4$ chance of satisfying the edge (each node gets color $c$ with probability $1/2$), whereas Algorithm 1 has a $1/3$ chance (the probability that $c$ is given higher global priority than the other two colors). This is precisely why Algorithm 1's approximation guarantee is a factor $4/3$ larger than the guarantee of Angel et al. (2016).

## 2.3. Graph MaxECC Algorithm

Although Algorithm 1 does not improve on the 0.3622-approximation of Ageev & Kononov (2020) for graph MAX-ECC, we can incorporate its distinguishing features (the color ordering $\pi$ and Observation 4) into a refined algorithm with a $154/405 \approx 0.38$ approximation factor. In order to improve on this extensively studied special case of the problem, our analysis is much more involved than the proof of Theorem 2.1 and requires proving several detailed technical lemmas that may be of interest in their own right. We provide an overview here and a full proof in Appendix B.

Our refined algorithm (Algorithm 2) for graphs solves the LP relaxation, generates color thresholds $\{\alpha_i : i \in [k]\}$, and generates a color ordering $\pi$ in the same way as Algorithm 1. It differs in that it partitions colors for each $v$ into colors that are *strong* or *weak* for $v$, given respectively by the sets

$$S_v = \{i \in [k] : x_v^i \geq 2/3\} \text{ and } W_v = [k] - S_v.$$

Since $\sum_{i=1}^k x_v^i = 1$, we know $|S_v| \in \{0, 1\}$. We say color $i$ is *strong for $v$* if $S_v = \{i\}$, otherwise $i$ is *weak for $v$*. Note that a color being *strong* or *weak* for $v$ is based on a fixed and non-random LP variable. This is separate from the notion of a color *wanting $v$*, which is a random event.

Algorithm 2 first checks if $v$ wants any weak colors. If so, it assigns $v$ the weak color of highest priority (using $\pi$) that $v$ wants. If $v$ wants no weak colors but has a strong color, then $v$ is assigned the strong color. Prioritizing weak colors in this way appears counterintuitive, since if $v$ has a strong color $c$ it suggests that $v$ should get color $c$. However, note that $x_v^c \geq 2/3$ still implies $v$ will get color $c$ with high probability, since a large value for $x_v^c$ makes it less likely $v$ will want any weak colors. Meanwhile, prioritizing weak colors enables us to lower bound the probability that an edge $e = (u, v)$ is satisfied even if $\ell(e)$ is weak for $u$ or $v$. We prove the following result for an arbitrary edge $e$, which by Observation 1 proves our approximation guarantee.

**Theorem 2.2.** *For every $e \in E$, $\mathbb{P}[e \text{ is satisfied}] \geq pz_e$ where $p = 154/405 > 0.3802$ when running Algorithm 2.*

*Proof sketch.* Fix $e = (u, v)$, let $c = \ell(e)$, and set $C = [k] \setminus \{c\}$. Define $C_v = \{i \in C : x_v^i \geq x_u^i\}$ and $C_u = C \setminus C_v$. As before, $T_e$ is the event that $e$ is satisfied and $X_v^i$ is the event that $v$ wants color $i$. Let $Y_v^i$ be the event that $v$ is *assigned* color $i$ by Algorithm 2, and $N_v$ be the event that $v$ wants *no* colors in $C$. If $c$ is strong for $v$, event $N_v$ is equivalent to event $Y_v^c$. Let $W_v'$ denote colors in $W_v$ that want $v$. Define $X_u^i, Y_u^i, W_u'$, and $N_u$ analogously for $u$. The proof is separated into (increasingly difficult) cases, based on how many of $\{u, v\}$ have $c$ as a strong color.

**Case 1: $S_u = S_v = \{c\}$.** Setting $W' = W_u' \cup W_v'$, we have

$$\mathbb{P}[T_e] = \prod_{i \in C} \mathbb{P}[i \notin W'] = \prod_{i \in C_v} \mathbb{P}[i \notin W_v'] \prod_{i \in C_u} \mathbb{P}[i \notin W_u']$$
$$= \prod_{i \in C_v}(1 - x_v^i) \prod_{i \in C_u}(1 - x_u^i) \geq \frac{2}{3} \cdot \frac{2}{3} \geq \frac{4}{9} z_e.$$

The first equality holds because $e$ is satisfied $\iff W_u' = W_v' = \emptyset$. The second follows from the definition of $C_u$ and $C_v$ and Observation 4. The second to last step can be shown using the constraint $\sum_{i=1}^k x_w^i = 1$ and the fact that $x_w^c \geq 2/3$ for $w \in \{u, v\}$ (see Lemma B.3 in Appendix B).

**Case 2: $S_v = \{c\}$, $c \in W_u$.** Satisfying $e$ is equivalent to event $Y_u^c \cap Y_v^c$, and $S_v = \{c\}$ implies $Y_v^c$ is equivalent to $N_v$. Note that $Y_u^c = Y_u^c \cap X_u^c$. In Appendix B we combine this with Bayes' Theorem, the independence of $X_u^c$ and $N_v$, and the fact that $P[X_u^c] = x_u^c \geq z_e$, to get

$$\mathbb{P}[T_e] = \mathbb{P}[Y_u^c \cap X_u^c \cap N_v] \geq \mathbb{P}[Y_u^c \mid N_v \cap X_u^c]\,\mathbb{P}[N_v]\,z_e.$$

Similar to Case 1, we use Lemma B.3 to prove that

$$\mathbb{P}[N_v] = \prod_{i \in C}\mathbb{P}[i \notin W_v'] = \prod_{i \in C}(1 - x_v^i) \geq \frac{2}{3}.$$

The most difficult steps for Case 2 are the two inequalities

$$\mathbb{P}[Y_u^c \mid N_v \cap X_u^c] \geq \mathbb{P}[Y_u^c \mid X_u^c] \geq 31/54,$$

where the first inequality relies on Lemma B.2 and the second on Lemma B.5, both of which require detailed proofs. Putting all these pieces together yields

$$\mathbb{P}\left[T_e\right] \geq \mathbb{P}\left[Y_u^c \mid X_u^c\right] \mathbb{P}\left[N_v\right] z_e \geq \frac{31}{54} \cdot \frac{2}{3} z_e = \frac{31}{81} z_e.$$

**Case 3:** $c \in W_u \cap W_u$. First condition on $Z_e = X_u^c \cap X_v^c$:

$$\mathbb{P}\left[T_e\right] = \mathbb{P}\left[T_e \mid Z_e\right] \mathbb{P}\left[Z_e\right] = \mathbb{P}\left[T_e \mid Z_e\right] z_e.$$

Define $D = (W_u \cup W_v) \backslash \{c\}$ to be the weak colors in $C$ that want at least one of $\{u, v\}$. The proof ultimately bounds $\mathbb{P}\left[T_e \mid Z_e\right]$, by conditioning on how many colors in $D$ want one or both of $\{u, v\}$. We partition $D$ into $D_u = \{i \in D: x_u^i \geq x_v^i\}$ and $D_v = D - D_u$, and using the same logic as Observation 4, note that $i \in D_u$ (respectively, $i \in D_v$) wants one or more nodes in $\{u, v\}$ if and only if $i$ wants $u$ (respectively, $i$ wants $v$). Let $D_u'$ be the colors in $D_u$ that want $u$ and define $D_v'$ analogously for $v$. Since $|D_u'|$ and $|D_v'|$ are independent, we have

$$\mathbb{P}[T_e \mid Z_e] = \sum_{i=0}^{|D_u|} \sum_{j=0}^{|D_v|} \frac{\mathbb{P}[|D_u'| = i] \, \mathbb{P}[|D_v'| = j]}{1 + i + j}, \quad (2)$$

where the fraction $1/(1 + i + j)$ is the probability that—conditioned on $i + j$ colors in $D$ wanting some nodes in $\{u, v\}$—the color $c$ will have higher priority than these $i + j$ other colors and hence $e$ will be satisfied. Using multiple applications of Lemma B.5 to bound sums of probabilities on the right hand side of Eq. (2), we show that $\mathbb{P}[T_e \mid Z_e] \geq 154/405$, which concludes the proof. □

# 3. Alternative ECC Objectives

We now turn our attention to more general ECC objectives, as well as several which incorporate notions of balance or fairness with respect to edge colors. Given space constraints, we defer proofs to Appendices C, D, and E.

## 3.1. Generalized Norm Objective

We begin by introducing $\ell_p$-NORM MINECC, which we define via the following convex program:

$$
\begin{aligned}
\min \quad & \left(\sum_{c=1}^k (m_c)^p\right)^{\frac{1}{p}} \\
\text{s.t.} \quad & \forall v \in V: & & \sum_{c=1}^k d_v^c \geq k - 1 \\
& \forall c \in [k], e \in E_c: & & d_v^c \leq \gamma_e \quad \forall v \in e \quad (3) \\
& \forall c \in [k]: & & m_c = \sum_{e \in E_c} \gamma_e \\
& d_v^c, \gamma_e \in \{0, 1\} & & \forall c \in [k], v \in V, e \in E.
\end{aligned}
$$

Recalling the notation of BLP (1), we think of $d_v^c = 1 - x_v^c$ as encoding the "distance" of node $v$ to color $c$, and $\gamma_e = 1 - z_e$ as encoding whether hyperedge $e$ is *unsatisfied* ($\gamma_e = 1$) or not ($\gamma_e = 0$). The variables $m_c$ can be thought of as the

entries in a per-color error vector $\mathbf{m} \in \mathbb{Z}^k$ which tracks the number of unsatisfied edges of each color. Otherwise, the variables and constraints are similar to BLP (1), adapted for the minimization objective. When $p = 1$, we recover the standard MINECC problem, i.e., the $\ell_1$-norm on $\mathbf{m}$. We now show how to approximate the generalized $\ell_p$-NORM MINECC problem. For $p \geq 1$, if we relax the binary constraints to linear constraints $0 \leq d_v^c \leq 1$ and $0 \leq \gamma_e \leq 1$, we obtain a convex programming relaxation. We can then apply convex optimization techniques to find the optimal solution to this relaxation and round the resulting fractional coloring. The rounding scheme is similar to that used for MINECC (Amburg et al., 2020). In particular, given a fractional coloring consisting of optimal variables $\{d_v^c\}$, we assign color $c$ to vertex $v$ if and only if $d_v^c < \frac{1}{2}$. We show that the resulting coloring is well-defined and a 2-approximation to the generalized objective. When $0 < p < 1$ we lose convexity, but we can recover approximations via reduction to SUBMODULAR VERTEX COVER.

**Theorem 3.1.** $\ell_p$-NORM MINECC *is approximable within factor* 2 *when* $p \geq 1$, *and factor* $2^{1/p}$ *when* $0 < p < 1$.

As $p$ increases, the $\ell_p$-norm on $\mathbf{m}$ converges to the maximum component value in $\mathbf{m}$. Thus, for $p \to \infty$ we recover a mini-max variant of ECC, for which the objective is to minimize the maximum number of unsatisfied edges of any color. This variant will be the focus of Section 3.2, but first we pause to note that this convergence of the $\ell_p$ objective implies that we cannot hope to obtain a better-than 2-approximation by rounding fractional solutions to Program (3).

**Theorem 3.2.** *As* $p \to \infty$, *the integrality gap of Program (3) converges to* 2.

## 3.2. Color-fair ECC variants

Now we focus on the mini-max case of $\ell_p$-NORM MINECC given by $p \to \infty$, which we refer to as COLOR-FAIR MINECC. In the following, $M_c^\lambda$ indicates the number of edges of color $c$ which are unsatisfied by a coloring $\lambda$.

---
COLOR-FAIR MINECC

**Input:** A $k$-edge-colored hypergraph $H = (V, E, \ell)$ and an integer $\tau$.

**Problem:** Does there exist a vertex coloring $\lambda$ of $H$ with $\max_{c \in [k]} M_c^\lambda \leq \tau$?

---

The optimization problem asks for a coloring $\lambda$ which minimizes $\max_{c \in [k]} M_c^\lambda$. Also of interest is the corresponding maxi-min variant COLOR-FAIR MAXECC, for which the question is whether there exists a coloring which satisfies at least $\tau$ edges of *every* color. These problems are not equivalent at optimality in general, though they are when restricted to hypergraphs with an equal number of edges of every color. We begin by establishing hardness for both problems.

**Theorem 3.3.** COLOR-FAIR MINECC *is* NP-*hard even when restricted to subcubic trees with cutwidth* 2*, exactly* 3 *edges of each color, and* $\tau = 2$*.* COLOR-FAIR MAXECC *is* NP-*hard even when restricted to paths with* $\tau = 1$*.*

Theorem 3.3 implies that it is NP-hard to provide any multiplicative approximation for COLOR-FAIR MAXECC, since it is NP-hard even to decide whether the optimum objective value is non-zero. Our reduction also rules out efficient algorithms in graphs of bounded feedback edge number, cutwidth, treewidth + max. degree, or slim tree-cut width. This highlights interesting differences between COLOR-FAIR MINECC and the standard ECC problem, which was shown to be FPT in terms of all four of these structural parameters (Kellerhals et al., 2023). On the positive side, we can give a FPT algorithm parameterized by the total number $t$ of unsatisfied edges. We remark (omitting the details) that the dependence on $t$ in the following theorem is tight under the exponential time hypothesis (ETH). To see this, we observe that the reduction of Theorem 3.3 creates linearly many vertices and edges with respect to the BOOLEAN SATISFIABILITY instance from which we reduce. In combination with a similar observation regarding the reduction of Tovey (1984), we conclude that under the ETH no $2^{o(|V|+|E|)}$-time (nor $2^{o(t)}$-time) algorithm exists for COLOR-FAIR MINECC.

**Theorem 3.4.** COLOR-FAIR MINECC *is solvable in* $O(2^t r |E|)$*-time.*

Though Theorems 3.3 and 3.4 close the most natural parameterized complexity questions, it remains open to determine whether COLOR-FAIR MINECC is hard when the number $k$ of colors is bounded. The standard ECC objective is in P when $k \leq 2$ and NP-hard whenever $k \geq 3$ (Amburg et al., 2020). We resolve this question via an intermediate hardness result for a VERTEX COVER variant which may be of independent interest. Specifically, in Appendix D we prove that given a bipartite graph $G = (A \uplus B, E)$ and an integer $\alpha$, it is NP-hard to determine whether there exists a vertex cover $C$ of $G$ with $\max\{|C \cap A|, |C \cap B|\} \leq \alpha$. This result facilitates the resolution of our question:

**Theorem 3.5.** COLOR-FAIR MINECC *and* COLOR-FAIR MAXECC *are* NP-*hard even in 2-regular hypergraphs with* $k = 2$*.*

We conclude this section by pointing the interested reader toward a few additional results. In Appendix D, we prove that an alternative to the 2-approximation implied by Theorem 3.1 can be obtained by reducing to a VERTEX COVER variant called SPARSE VERTEX COVER and applying a result of Blum et al. (2022). A natural question is whether COLOR-FAIR MINECC is approximable below factor 2. Any such algorithm is likely to require new techniques. Theorem 3.2 implies that the first of our two strategies cannot be improved, while any improvement in the existing

2-approximation for SPARSE VERTEX COVER would refute the unique games conjecture (Blum et al., 2022).

It is also compelling to develop purely combinatorial approaches, i.e., algorithms which do not rely upon convex optimization. As a first step, in Appendix D we show how to analyze an existing linear-time and purely combinatorial ECC algorithm of Veldt (2023), proving that it yields a $k$-approximation to the COLOR-FAIR MINECC objective.

### 3.3. Protected-color ECC

COLOR-FAIR MINECC encodes fairness by ensuring that no single color is heavily "punished" (as measured by unsatisfied hyperedges), but what if we are only concerned about a single color that represents a protected interaction type? It is always trivially possible to ensure that every edge of a given color $c_1$ is satisfied by setting $\lambda(v) = c_1$ for every vertex $v$. This strategy, however, completely disregards all other edges. We might reasonably assume that while we are constrained by concern for a given protected color, we still wish to find a high-quality solution as measured by the traditional clustering objective. This tension is captured in the following problem definition.

---

PROTECTED-COLOR MINECC

**Input:** An edge-colored hypergraph $H = (V, E, \ell)$, two integers $t, b$, and a protected color $c_1$.

**Problem:** Is it possible to color $V$ such that at most $t$ edges are unsatisfied, of which at most $b$ have color $c_1$?

---

When $t = b$ we recover the standard ECC problem, so PROTECTED-COLOR MINECC is NP-hard. However, by rounding fractional solutions to the following linear program we recover bicriteria $(\alpha, \beta)$-approximations. If $\mathsf{opt}_b$ is the minimum possible number of unsatisfied edges subject to the constraint $b$, then these algorithms guarantee at most $\alpha \cdot \mathsf{opt}_b$ unsatisfied edges, of which at most $\beta \cdot b$ have color $c_1$.

$$
\begin{aligned}
\min \quad & \sum_{e \in E} \gamma_e \\
\text{s.t.} \quad & \forall v \in V : && \sum_{c=1}^{k} d_v^c \geq k - 1 \\
& \forall c \in [k], e \in E_c : && d_v^c \leq \gamma_e \quad \forall v \in e && (4) \\
& \sum_{e \in E_{c_1}} \gamma_e \leq b \\
& d_v^c, \gamma_e \in [0, 1] && \forall c \in [k], v \in V, e \in E
\end{aligned}
$$

All variables have the same meaning as in Program (3), but we have added the third constraint which encodes that at most $b$ edges of our protected color $c_1$ may be unsatisfied. Apart from this constraint, LP (4) is identical to the canonical MINECC LP used to obtain state-of-the-art approxima-

*Table 1.* We report hypergraph statistics, and compare the standard (ST) MINECC algorithm against the COLOR-FAIR MINECC (CF) algorithm. Runtimes are given in seconds taken to solve the LP. Objective numbers are an upper bound on the approximation ratio, given by the objective value for the algorithm divided by the lower bound for the objective determined by the LP relaxation.

| Dataset | $|V|$ | $|E|$ | $r$ | $k$ | Runtime (s) | | ECC Obj | | CFECC Obj | |
|---|---|---|---|---|---|---|---|---|---|---|
| | | | | | ST | CF | ST | CF | ST | CF |
| *Brain* | 638 | 21180 | 2 | 2 | 0.50 | 1.57 | 1.00 | 1.02 | 1.26 | 1.00 |
| *Cooking* | 6714 | 39774 | 65 | 20 | 46.76 | 52.88 | 1.00 | 1.00 | 1.66 | 1.66 |
| *DAWN* | 2109 | 87104 | 22 | 10 | 4.49 | 11.60 | 1.00 | 1.00 | 1.39 | 1.38 |
| *MAG-10* | 80198 | 51889 | 25 | 10 | 8.40 | 15.52 | 1.00 | 1.20 | 1.48 | 1.03 |
| *Trivago-Clickout* | 207974 | 247362 | 85 | 55 | 116.94 | 129.55 | 1.00 | 1.37 | 1.68 | 1.01 |
| *Walmart-Trips* | 88837 | 65898 | 25 | 44 | 156.71 | 314.03 | 1.00 | 1.05 | 2.48 | 1.56 |

tions (Amburg et al., 2020; Veldt, 2023), and our algorithm can be seen as an appropriate adaption of those rounding schemes. In the following, $\rho \in (0, \frac{1}{2}]$ is a parameter which allows us to tune the approximation guarantees $\alpha$ and $\beta$.

**Theorem 3.6.** *For every* $\rho \in (0, \frac{1}{2}]$*, there exists a polynomial-time* $(\frac{1}{\rho}, \frac{1}{1-\rho})$*-approximation for* PROTECTED-COLOR MINECC.

We also give a FPT algorithm.

**Theorem 3.7.** PROTECTED-COLOR MINECC *is FPT with respect to the total number* $t$ *of unsatisfied edges.*

Given these positive results, a natural question is whether we may efficiently solve the MINECC objective with *multiple* protected colors. We conclude by showing that as soon as we allow more than one protected color, it becomes hard even to determine whether any solution exists.

**Theorem 3.8.** *Given a* 2*-regular* $H = (V, E, \ell)$*, two integers* $b_1, b_2$*, and two colors* $c_1, c_2$*, it is* NP*-hard to determine whether it is possible to color* $V$ *such that at most* $b_1$ *edges of color* $c_1$ *and at most* $b_2$ *edges of color* $c_2$ *are unsatisfied.*

## 4. Experiments

While we primarily focus on theoretical results, we also implemented and evaluated the performance of our alternate objective ECC algorithms on a standard suite of benchmark hypergraphs. LP-based methods for standard ECC are known to perform far better in practice than their theoretical guarantees, often yielding optimal solutions (Amburg et al., 2020). Thus, our MAXECC algorithms should be viewed as theoretical results that help bridge the theory-practice gap and are thus not the focus of our experiments. Code for our experiments can be found at https://github.com/tommy1019/AltECC.

**Datasets.** We consider edge-colored hypergraphs from Amburg et al. (2020) and Veldt (2023). *Brain* is derived from brain MRI data (Crossley et al., 2013); nodes represent

brain regions, edges of one color indicate regions with high fMRI correlation, and edges of the other color indicate regions with similar activation patterns. *Cooking* is derived from the What's Cooking dataset (Kaggle, 2015a); nodes are food ingredients, hyperedges are ingredients encoding a recipe, and colors indicate cuisine (e.g., Indian or Korean). In *DAWN*, nodes are drugs from the Drug Abuse Warning Network (Substance Abuse and Mental Health Services Administration.), and hyperedges are groups of drugs ingested by an individual prior to a trip to the emergency room. Colors indicate patient outcome (e.g., "sent home" or "surgery"). *MAG-10* is derived from publication data from the Microsoft Academic Graph (Sinha et al., 2015); nodes are academic authors, hyperedges indicate authors on a publication, and hyperedge colors indicate the publication venue. *Walmart-Trips* is derived from the Kaggle trip type classification data (Kaggle, 2015b); nodes are products sold at Walmart, and hyperedges represents sets of products purchased in the same shopping trip. Colors are shopping trip types. *Trivago-Clickout* is derived from the 2019 ACM RecSys Challenge data (Knees et al., 2019); nodes are vacation rentals on Trivago, hyperedges are sets of rentals that a user "clicks out" on during the same browsing session. Colors encode the country location where the browsing session occurred (e.g., Spain if the browsing happened on Trivago.es). Table 1 provides statistics for each hypergraph.

**Implementation.** We refer to our LP algorithms for COLOR-FAIR MINECC and PROTECTED-COLOR MINECC as CF and PC respectively, and refer the standard MINECC LP rounding algorithm as ST. These were implemented in the Julia programming language and run on a research server running Ubuntu 20.04.1 with two AMD EPYC 7543 32-Core Processors and 1 TB of RAM. Our implementation of these algorithms chooses a color that has the lowest LP distance variable among all colors for each node. This approach can only improve our approximation guarantees over the theoretical approach that only assigns a color if there is an LP variable that is less than $1/2$. Our implementation for PC then performs at least as well as the algorithm from

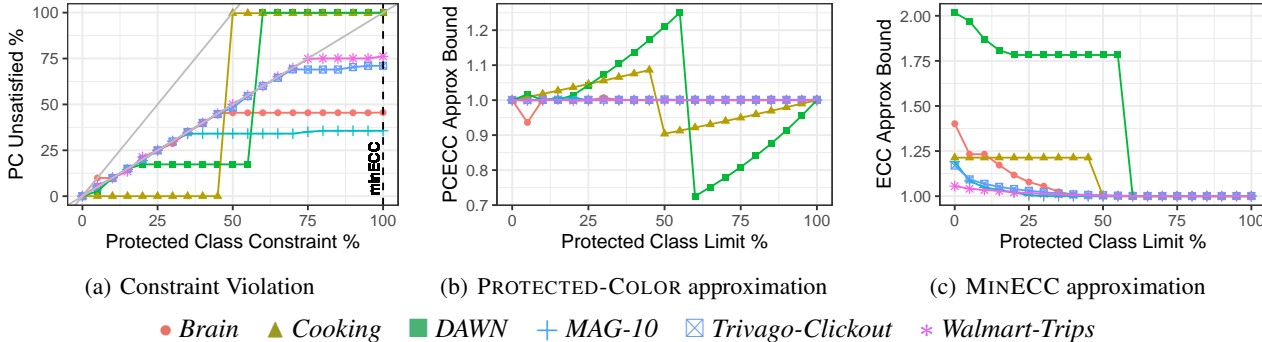

*Figure 2.* Results for PC while varying the size of the constraint $b$, given as a percent of the total number of protected edges. The protected color is always chosen to be the color with the median number of edges among all colors in the hypergraph. Figure (a) shows the percent of protected edges left unsatisfied by PC, versus the maximum percent of edges that the constraint allows to be unsatisfied. Line $y = x$ represents the constraint imposed by the problem definition, and $y = 2x$ is the theoretical limit for our bicriteria approximation. Figures (b) and (c) are upper bounds on the approximation ratio achieved by PC for the protected color objective and for MINECC respectively.

Theorem 3.6 with $\rho = 1/2$, so it is a $(2, 2)$-approximation. We used Gurobi optimization software with default settings to solve LPs. The bottleneck for all algorithms is solving the LP, which takes between few seconds and a few minutes depending on the dataset. The time to round the LP is negligible (under 0.3 seconds in the worst case).

**COLOR-FAIR MINECC**. Table 1 compares CF against ST. We first observe that in practice, CF achieves approximation factors that are much better than the theoretical bound of 2, and in several cases are very close to 1. Furthermore, CF even does well approximating the standard MINECC objective even though it is not directly designed for it. In the worst case, it achieves a 1.37-approximation, which is only 37% worse than ST, which finds an optimal solution for MINECC. By comparison, ST it is up to 66% worse than CF for the color-fair objective. In fact, on average among the datasets we tested, CF is only 10.66% worse than ST on the standard MINECC objective, while ST is 32.62% worse than CF on the color-fair objective. In other words, the cost of incorporating fairness into the ECC framework is not too high—CF maintains good performance in terms of the MINECC objective while showing significant improvements for the color-fair objective.

**PROTECTED-COLOR MINECC**. Figure 2(a) shows the constraint satisfaction for PC as $b$ (the number of unsatisfied protected edges) varies. Again we see that our algorithm tends to far exceed theoretical guarantees. In most cases, the number of protected edges left unsatisfied by PC is below the constraint $b$, even though the algorithm can in theory violate this constraint by up to a factor 2. Results are especially good for *Brain*, *MAG-10*, *Walmart*, and *Trivago-Clickout*, where the constraint tends to be satisfied and the objective value for CF is within a factor 1.007 of the LP lower bound (Figure 2(b)). PC also tends to satisfy the protected color constraint for *DAWN* and *Cooking* when $b$ is below 50%

of the total number of protected edges, achieving a 1.3-approximation or better in this case (Figure 2(b)). Once $b$ is above this 50% threshold, PC satisfies no protected edges on *DAWN* and *Cooking*. This still matches our theory: in order to *guarantee* (based on our theory) we do not leave all protected edges unsatisfied, $b$ must be less than 50% of these edges. PC also does a good job of approximating the standard MINECC objective (factor 2 or better, see Figure 2(c)), while incorporating this protected color constraint.

We note that the rightmost point in Figure 2(a) corresponds to standard MINECC as there is no bound on unsatisfied protected edges. From this, we can examine how badly ST violates protected color constraints for small $b$ values. On *MAG-10* when $b$ is 10% of the number of protected edges, ST violates the constraint by a factor of 3.60 whereas PC only violates the constraint by a factor of 1.01. This gap is even more pronounced for the *DAWN* dataset at 10%, where ST violates the constraint by a factor of 10.3 while PC satisfies the constraint. Thus, for situations where there is a need to protect certain edge types, standard MINECC algorithms do not provide meaningful solutions.

## 5. Conclusions

We have established the first approximation algorithm for hypergraph MAXECC and improved the best known approximation factor for graph MAXECC. We also introduced two ECC variants that incorporate fairness, designing new approximation algorithms which in practice far exceed their theoretical guarantees. One open direction is to try to further improve the best approximation ratio for MAXECC. This may require deviating significantly from previous LP rounding techniques, since it is currently very challenging to improve the approximation by even a small amount using this approach. Another open direction is to improve on the $2^{1/p}$-approximation for $\ell_p$-NORM MINECC when $p < 1$.

## Acknowledgements

This work was supported in part by the Gordon & Betty Moore Foundation under award GBMF4560 to Blair D. Sullivan, by the National Science Foundation under award IIS-1956286 to Blair D. Sullivan, and by the Army Research Office under award W911NF-24-1-0156 to Nate Veldt.

## Impact Statement

This paper presents work whose goal is to advance the field of Machine Learning. There are many potential societal consequences of our work, none of which we feel must be specifically highlighted here.

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

## A. Parameterized Complexity

For a complete introduction to parameterized complexity, we refer the reader to the standard text (Cygan et al., 2015). A *parameterized problem* is a language $L \subseteq \Sigma^*$, where $\Sigma$ is a fixed, finite alphabet. An *instance* is a tuple $(I, k) \in \Sigma^* \times \mathbb{N}$, where $k$ is the *parameter*. A parameterized problem is called *fixed-parameter tractable* (FPT) if there exists an algorithm $\mathcal{A}$, a computable function $f \colon \mathbb{N} \to \mathbb{N}$ and a constant $c$ such that for every instance $(I, k)$, $\mathcal{A}$ decides whether $(I, k) \in L$ in time bounded by $f(k) \cdot |(I, k)|^c$. This means the algorithm $\mathcal{A}$ has polynomial dependence on the instance size, while the exponential (or worse) part of the running time is dependent only on the parameter $k$.

The standard ECC problem is FPT with respect to both *natural* parameters, i.e., the number of unsatisfied edges and the number of satisfied edges (Cai & Leung, 2018), as well as several *structural* parameters, notably the slim tree-cut width, feedback edge number, cutwidth, and treewidth plus maximum degree (Kellerhals et al., 2023). Meanwhile, it is (under standard complexity assumptions) not FPT with respect to vertex-cover number or treewidth (Kellerhals et al., 2023).

## B. MAXECC **Proofs**

We use the following lemma as a small step in our approximation guarantee for hypergraph MAXECC (Theorem 2.1). The result was originally used in designing a $1/e^2$-approximation algorithm for graph MAXECC (Angel et al., 2016). A full proof can be found in the work of (Angel et al., 2016).

**Lemma B.1** (Lemma 4 of Angel et al. (2016))**.** *Let $\{X_1, X_2, \ldots, X_j\}$ be a set of independent events satisfying $\sum_{i=1}^{j} \mathbb{P}[X_i] \leq 1$, then the probability that at most one of them happens is greater than or equal to $2/e$.*

The remainder of our supporting lemmas are new results that we prove for our approximation algorithm for graph MAXECC.

**Lemma B.2.** *Given an edge $e = (u, v)$ of color $c$ with $c \in W_u$, when running Algorithm 2 we have*

$$\mathbb{P}\left[Y_u^c \mid N_v \cap X_u^c\right] \geq \mathbb{P}\left[Y_u^c \mid X_u^c\right].$$

*Proof.* For a node $u$ and color $i$ we use $\overline{X}_u^c$ to denote the event that $u$ does not want $c$.

We first observe that it is sufficient to show positive correlation between $Y_u^c$ and $N_v$. To see this, we use Bayes' Theorem and the independence of $X_u^c$ and $N_v$ to write

$$\mathbb{P}\left[Y_u^c \mid N_v \cap X_u^c\right] = \frac{\mathbb{P}\left[Y_u^c \cap N_v \cap X_u^c\right]}{\mathbb{P}\left[N_v \cap X_u^c\right]} = \frac{\mathbb{P}\left[N_v \cap X_u^c \mid Y_u^c\right] \mathbb{P}\left[Y_u^c\right]}{\mathbb{P}\left[N_v \cap X_u^c\right]} = \frac{\mathbb{P}\left[N_v \cap X_u^c \mid Y_u^c\right] \mathbb{P}\left[Y_u^c\right]}{\mathbb{P}\left[N_v\right] \cdot \mathbb{P}\left[X_u^c\right]},$$

as well as

$$\mathbb{P}\left[Y_u^c \mid X_u^c\right] = \frac{\mathbb{P}\left[X_u^c \cap Y_u^c\right]}{\mathbb{P}\left[X_u^c\right]} = \frac{\mathbb{P}\left[X_u^c \mid Y_u^c\right] \mathbb{P}\left[Y_u^c\right]}{\mathbb{P}\left[X_u^c\right]} = \frac{1 \cdot \mathbb{P}\left[Y_u^c\right]}{\mathbb{P}\left[X_u^c\right]}.$$

By rearranging terms, we see that our claim is equivalent to

$$\mathbb{P}\left[N_v \cap X_u^c \mid Y_u^c\right] = \mathbb{P}\left[N_v \mid Y_u^c\right] \geq \mathbb{P}\left[N_v\right].$$

We complete the proof by demonstrating the equivalent inequality $\mathbb{P}\left[Y_u^c \mid N_v\right] \geq \mathbb{P}\left[Y_u^c\right]$. Impose an arbitrary order $c_1, c_2 \ldots, c_k$ on the color set, and without loss of generality assume that $\ell(e) = c = c_k$. In what follows, we always write $c$ for $c_k$ and whenever considering a subscripted color $c_i$ we assume that $i \in W_u \setminus \{c\}$.

If node $u$ has a strong color, then without loss of generality we assume that the strong color is $c_{k-1}$. The remainder of the proof is written for the case where $S_u = \emptyset$, in which case the set of weak colors (other than $c$) that might want node $u$ is $W_u \setminus \{c\} = \{c_1, c_2, \ldots, c_{k-1}\}$. In other words, $W_u$ is the set of colors with indices in $[k-1]$. If instead we had $S_u = \{c_{k-1}\}$ and $W_u \setminus \{c\} = \{c_1, c_2, \ldots, c_{k-2}\}$, the proof works in exactly the same way if we instead considered the set of colors associated with indices in $[k-2]$ instead of indices in $[k-1]$.

Now, consider all possible subsets of $[k-1]$ which may define (according to our ordering) those weak colors (besides $c$) which are wanted by $u$. For a particular subset $S \subseteq [k-1]$, we write $X_u^S$ for the event that $u$ wants exactly (not considering $c$) the colors in $S$. In the following, we write $w(S) = \frac{1}{|S|+1}$ and call this quantity the *weight* of $S$. The events $X_u^S$ partition the sample space, so we may write

$$\mathbb{P}\left[Y_u^c\right] = \sum_{S \subseteq [k-1]} \mathbb{P}\left[Y_u^c \mid X_u^S\right] \cdot \mathbb{P}\left[X_u^S\right] = \mathbb{P}\left[X_u^c\right] \cdot \sum_{S \subseteq [k-1]} w(S) \cdot \mathbb{P}\left[X_u^S\right].$$

In the above, the second equality follows from the observation that, conditioned on $X_u^S$ for any $S \subseteq [k-1]$, the event $Y_u^c$ occurs if and only if (a) $u$ wants $c$ and (b) $c$ preceeds each color in $S$ in the global ordering of colors. The conditions (a) and (b) are independent, and occur with probabilities $\mathbb{P}[X_u^c]$ and $w(S)$, respectively. We will now make use of the independence of the $X_u^{c_i}$ events to write $\mathbb{P}[X_u^S]$ as the product of $k-1$ probabilities.

$$\mathbb{P}[Y_u^c] = \mathbb{P}[X_u^c] \cdot \sum_{S \subseteq [k-1]} \left[ w(S) \cdot \left( \prod_{i \in S} \mathbb{P}[X_u^{c_i}] \right) \cdot \left( \prod_{j \notin S} \mathbb{P}\left[ \overline{X}_u^{c_j} \right] \right) \right]. \tag{5}$$

We want to write a similar expression for $\mathbb{P}[Y_u^c | N_v]$. We begin by using similar reasoning as above to write

$$\mathbb{P}[Y_u^c \mid N_v] = \sum_{S \subseteq [k-1]} \mathbb{P}\left[ Y_u^c \mid X_u^S \cap N_v \right] \cdot \mathbb{P}\left[ X_u^S \mid N_v \right]$$

$$= \mathbb{P}[X_u^c \mid N_v] \cdot \sum_{S \subseteq [k-1]} w(S) \cdot \mathbb{P}\left[ X_u^S \mid N_v \right]$$

$$= \mathbb{P}[X_u^c] \cdot \sum_{S \subseteq [k-1]} w(S) \cdot \mathbb{P}\left[ X_u^S \mid N_v \right].$$

Next, we observe that the $X_u^{c_i}$ events remain independent even when conditioned on $N_v$. This allows us to write

$$\mathbb{P}[Y_u^c \mid N_v] = \mathbb{P}[X_u^c] \cdot \sum_{S \subseteq [k-1]} \left[ w(S) \cdot \left( \prod_{i \in S} \mathbb{P}[X_u^{c_i} \mid N_v] \right) \cdot \left( \prod_{j \notin S} \mathbb{P}\left[ \overline{X}_u^{c_j} \mid N_v \right] \right) \right].$$

To simplify this expression further, we claim that for every $i \in [k-1]$, $\mathbb{P}[X_u^{c_i} \mid N_v] = \mathbb{P}\left[ X_u^{c_i} \mid \overline{X}_v^{c_i} \right]$. This can be derived from the observation that $\bigcap_{j \neq i} \overline{X}_v^{c_j}$ is independent of $X_u^{c_i}$ when conditioned on $\overline{X}_v^{c_i}$, as well as being unconditionally independent of $X_v^{c_i}$. We then manipulate the definition of conditional probability to see that

$$\mathbb{P}[X_u^{c_i} \mid N_v] = \frac{\mathbb{P}\left[ X_u^{c_i} \cap \bigcap_{j \neq i} \overline{X}_v^{c_j} \cap \overline{X}_v^{c_i} \right]}{\mathbb{P}\left[ \bigcap_{j \neq i} \overline{X}_v^{c_j} \cap \overline{X}_v^{c_i} \right]}$$

$$= \frac{\mathbb{P}\left[ X_u^{c_i} \cap \bigcap_{j \neq i} \overline{X}_v^{c_j} \cap \overline{X}_v^{c_i} \right]}{\mathbb{P}\left[ \bigcap_{j \neq i} \overline{X}_v^{c_j} \right] \cdot \mathbb{P}\left[ \overline{X}_v^{c_i} \right]}$$

$$= \frac{\mathbb{P}\left[ X_u^{c_i} \cap \bigcap_{j \neq i} \overline{X}_v^{c_j} \mid \overline{X}_v^{c_i} \right]}{\mathbb{P}\left[ \bigcap_{j \neq i} \overline{X}_v^{c_j} \right]}$$

$$= \frac{\mathbb{P}\left[ X_u^{c_i} \mid \overline{X}_v^{c_i} \right] \cdot \mathbb{P}\left[ \bigcap_{j \neq i} \overline{X}_v^{c_j} \mid \overline{X}_v^{c_i} \right]}{\mathbb{P}\left[ \bigcap_{j \neq i} \overline{X}_v^{c_j} \right]}$$

$$= \frac{\mathbb{P}\left[ X_u^{c_i} \mid \overline{X}_v^{c_i} \right] \cdot \mathbb{P}\left[ \bigcap_{j \neq i} \overline{X}_v^{c_j} \right]}{\mathbb{P}\left[ \bigcap_{j \neq i} \overline{X}_v^{c_j} \right]}$$

$$= \mathbb{P}\left[ X_u^{c_i} \mid \overline{X}_v^{c_i} \right].$$

We can now write a suitable counterpart to Equation (5):

$$\mathbb{P}[Y_u^c \mid N_v] = \mathbb{P}[X_u^c] \cdot \sum_{S \subseteq [k-1]} \left[ w(S) \cdot \left( \prod_{i \in S} \mathbb{P}\left[ X_u^{c_i} \mid \overline{X}_v^{c_i} \right] \right) \cdot \left( \prod_{j \notin S} \mathbb{P}\left[ \overline{X}_u^{c_j} \mid \overline{X}_v^{c_j} \right] \right) \right]. \tag{6}$$

We will complete the proof by showing that the RHS of Equation (5) is a lower bound for the RHS of Equation (6). We accomplish this in $k-1$ steps, one for each color besides $c$. In the first step, we begin by rearranging the terms of Equation (6) to isolate $\mathbb{P}\left[X_u^{c_1} \mid \overline{X}_v^{c_1}\right]$ and $\mathbb{P}\left[\overline{X}_u^{c_1} \mid \overline{X}_v^{c_1}\right]$:

$$
\begin{aligned}
\mathbb{P}\left[Y_u^c \mid N_v\right] = \mathbb{P}\left[X_u^c\right] \cdot \Bigg[ & \mathbb{P}\left[X_u^{c_1} \mid \overline{X}_v^{c_1}\right] \cdot \sum_{S \ni 1} w(S) \cdot \left(\prod_{1 \neq i \in S} \mathbb{P}\left[X_u^{c_i} \mid \overline{X}_v^{c_i}\right]\right) \cdot \left(\prod_{j \notin S} \mathbb{P}\left[\overline{X}_u^{c_j} \mid \overline{X}_v^{c_j}\right]\right) \\
& + \mathbb{P}\left[\overline{X}_u^{c_1} \mid \overline{X}_v^{c_1}\right] \cdot \sum_{S \not\ni 1} w(S) \cdot \left(\prod_{i \in S} \mathbb{P}\left[X_u^{c_i} \mid \overline{X}_v^{c_i}\right]\right) \cdot \left(\prod_{1 \neq j \notin S} \mathbb{P}\left[\overline{X}_u^{c_j} \mid \overline{X}_v^{c_j}\right]\right) \Bigg] .
\end{aligned}
$$

In the above, we refer to the first and second sums as the $c_1$-*wanted coefficient* and the $c_1$-*not-wanted coefficient*, respectively. Observe that there exists a natural bijection between the summands in these two coefficients. Specifically, we map the summand in the $c_1$-wanted coefficient corresponding to set $S$ to the summand in the $c_1$-not-wanted coefficient corresponding to the set $S \setminus \{1\}$. These two summands are identical up to the difference between $w(S)$ and $w(S \setminus \{1\})$. By definition, the former weight is smaller. Hence, the $c_1$-wanted coefficient is smaller than the $c_1$-not-wanted coefficient.

Next, we claim that $\mathbb{P}\left[X_u^{c_1} \mid \overline{X}_v^{c_1}\right] \leq \mathbb{P}[X_u^{c_1}]$, and (equivalently) $\mathbb{P}\left[\overline{X}_u^{c_1} \mid \overline{X}_v^{c_1}\right] \geq \mathbb{P}\left[\overline{X}_u^{c_1}\right]$. To prove this claim, we consider two cases. If $x_u^{c_1} \leq x_v^{c_1}$, then $\mathbb{P}\left[X_u^{c_1} \mid \overline{X}_v^{c_1}\right] = 0 \leq x_u^{c_1} = \mathbb{P}[X_u^{c_1}]$. Otherwise, $x_u^{c_1} > x_v^{c_1}$, and we have $\mathbb{P}\left[X_u^{c_1} \mid \overline{X}_v^{c_1}\right] = \frac{x_u^{c_1} - x_v^{c_1}}{1 - x_v^{c_1}} = x_u^{c_1} \frac{1 - x_v^{c_1}/x_u^{c_1}}{1 - x_v^{c_1}} \leq x_u^{c_1} = \mathbb{P}[X_u^{c_1}]$.

We now rewrite our expression for $\mathbb{P}\left[Y_u^c \mid N_v\right]$ in terms of *unconditional* probabilities of $u$ (not) wanting color $c_1$. In the following, let $0 \leq d = \mathbb{P}\left[X_u^{c_1}\right] - \mathbb{P}\left[X_u^{c_1} \mid \overline{X}_v^{c_1}\right]$.

$$
\begin{aligned}
\mathbb{P}\left[Y_u^c \mid N_v\right] = \mathbb{P}\left[X_u^c\right] \cdot \Bigg[ & \left(\mathbb{P}\left[X_u^{c_1}\right] - d\right) \cdot \sum_{S \ni 1} w(S) \cdot \left(\prod_{1 \neq i \in S} \mathbb{P}\left[X_u^{c_i} \mid \overline{X}_v^{c_i}\right]\right) \cdot \left(\prod_{j \notin S} \mathbb{P}\left[\overline{X}_u^{c_j} \mid \overline{X}_v^{c_j}\right]\right) \\
& + \left(\mathbb{P}\left[\overline{X}_u^{c_1}\right] + d\right) \cdot \sum_{S \not\ni 1} w(S) \cdot \left(\prod_{i \in S} \mathbb{P}\left[X_u^{c_i} \mid \overline{X}_v^{c_i}\right]\right) \cdot \left(\prod_{1 \neq j \notin S} \mathbb{P}\left[\overline{X}_u^{c_j} \mid \overline{X}_v^{c_j}\right]\right) \Bigg] .
\end{aligned}
$$

Our observation that the $c_1$-wanted coefficient is smaller than the $c_1$-not-wanted coefficient yields the bound we desire:

$$
\begin{aligned}
\mathbb{P}\left[Y_u^c \mid N_v\right] \geq \mathbb{P}\left[X_u^c\right] \cdot \Bigg[ & \mathbb{P}\left[X_u^{c_1}\right] \cdot \sum_{S \ni 1} w(S) \cdot \left(\prod_{1 \neq i \in S} \mathbb{P}\left[X_u^{c_i} \mid \overline{X}_v^{c_i}\right]\right) \cdot \left(\prod_{j \notin S} \mathbb{P}\left[\overline{X}_u^{c_j} \mid \overline{X}_v^{c_j}\right]\right) \\
& + \mathbb{P}\left[\overline{X}_u^{c_1}\right] \cdot \sum_{S \not\ni 1} w(S) \cdot \left(\prod_{i \in S} \mathbb{P}\left[X_u^{c_i} \mid \overline{X}_v^{c_i}\right]\right) \cdot \left(\prod_{1 \neq j \notin S} \mathbb{P}\left[\overline{X}_u^{c_j} \mid \overline{X}_v^{c_j}\right]\right) \Bigg] .
\end{aligned}
$$

This completes the first of $k-1$ steps. In the next step, we begin by rearranging terms in the inequality above to isolate $\mathbb{P}\left[X_u^{c_2} \mid \overline{X}_v^{c_2}\right]$ and $\mathbb{P}\left[\overline{X}_u^{c_2} \mid \overline{X}_v^{c_2}\right]$. We then apply the same analysis, yielding another lower bound on $\mathbb{P}\left[Y_u^c \mid N_v\right]$, this time written in terms of unconditional probabilities of $u$ (not) wanting colors $c_1$ and $c_2$, and conditional probabilities of $u$ (not) wanting colors $c_3, c_4, \ldots c_{k-1}$. After $k-1$ steps, our lower bound becomes identical (up to rearranging terms) to the RHS of Equation 5, as desired. $\square$

**Lemma B.3.** *Let $\beta \in [0, 1]$ be a constant and $\mathbf{x} \in \mathbb{R}_{\geq 0}^m$ be a length $m$ nonnegative vector satisfying $\sum_{t=1}^m x_t \leq \beta$, then we have $\prod_{t=1}^m (1 - x_t) \geq 1 - \beta$.*

*Proof.* Observe that the claim is trivially true when $m = 1$. We proceed via induction on $m$. For $m > 1$, we have $\sum_{t=1}^{m-1} \leq \beta - x_m$, so by the inductive hypothesis

$$\prod_{t=1}^m (1 - x_t) = (1 - x_m) \cdot \prod_{t=1}^{m-1} (1 - x_t) \geq (1 - x_m)(1 - \beta + x_m) = 1 - \beta + x_m(\beta - x_m) \geq 1 - \beta.$$

$\square$

Before presenting our next lemma we first require the following definition.

**Definition B.4.** Let $m$ be a nonnegative integer and $t$ be an integer. Given a vector $\mathbf{x} \in \mathbb{R}^m$, if $m \geq t > 0$, we define the function $P(\mathbf{x}, t)$ as follows:

$$P(\mathbf{x}, t) = \sum_{I \in \binom{[m]}{t}} \left( \prod_{i \in I} x_i \prod_{j \in [m] \setminus I} (1 - x_j) \right),$$

and for other choices of $t$ and $m$ we define

$$P(\mathbf{x}, t) = \begin{cases} 0 & \text{if } m < t \text{ or } t < 0 \\ 1 & \text{if } 0 = m = t \\ \prod_{i=1}^m (1 - x_i) & \text{if } 0 = t < m. \end{cases}$$

To provide intuition, $P(\mathbf{x}, t)$ is defined to encode the probability that $t$ events from a set of $m$ independent events happen. In more detail, consider a set of $m$ mutually independent events $\mathcal{X} = \{X_1, X_2, \cdots, X_m\}$, where the probability that the $i$th event happens is $x_i = \mathbb{P}[X_i]$, the $i$th entry of $\mathbf{x}$. The function $P$ in Definition B.4 is the probability that exactly $t$ of the events in $\mathcal{X}$ happen. In our proofs for MAXECC, we will apply this with $\mathcal{X}$ representing a subset of colors in the ECC instance, while the vector $\mathbf{x}$ encodes LP variables $\{x_u^i\}$ for some node $u$ and that set of colors (which by Observation 2 are probabilities for wanting those colors). Lemmas B.5 and B.6 will aid in bounding the probability that a node is assigned a certain color, conditioned on how many other colors it wants.

**Lemma B.5.** *Let $m \geq 2$ be an integer and $a_0, a_1, \ldots, a_m$ be values such that for every $t \in [0, m - 2]$*

$$a_{t+1} \leq a_t$$
$$2a_{t+1} \leq a_t + a_{t+2}.$$

*Let $\mathcal{D} = \{\mathbf{x} \in [0, 2/3]^m : \sum_{i=1}^m x_i \leq 1\}$ be the domain of a function $f : \mathcal{D} \to \mathbb{R}$ defined as*

$$f(\mathbf{x}) = \sum_{t=0}^m a_t P(\mathbf{x}, t).$$

*Then $f$ is minimized (over domain $\mathcal{D}$) by any vector $\mathbf{x}^*$ with one entry set to $2/3$, one entry set to $1/3$, and every other entry set to 0. Furthermore we have*

$$f(\mathbf{x}^*) = \frac{2}{9}(a_0 + a_2) + \frac{5}{9}a_1. \tag{7}$$

*Proof.* Let $\mathcal{S} = \{\mathbf{x} \in [0, 2/3]^m : \sum_{i=1}^m x_i = 1\} \subsetneq \mathcal{D}$ be the set of input vectors for $f$ whose entries sum exactly to 1. Let $\mathcal{I} \subsetneq \mathcal{S}$ be the set of vectors in $\mathcal{S}$ with one entry equal to $2/3$, one entry equal to $1/3$, and all other entries equal to 0. More formally, if $\mathbf{x} \in \mathcal{I}$, this means there exists two distinct indices $\{i, j\}$ such that $x_i = 2/3$, $x_j = 1/3$, and $x_k = 0$ for every $k \notin \{i, j\}$. Our goal is to prove there exists some $\mathbf{x} \in \mathcal{I}$ that minimizes $f$. We prove this in two steps: (1) we show that there exists a minimizer of $f$ in $\mathcal{S}$, and then (2) we show how to convert an arbitrary vector $\mathbf{x} \in \mathcal{S}$ into a new vector $\mathbf{x}^* \in \mathcal{I}$ satisfying $f(\mathbf{x}^*) \leq f(\mathbf{x})$. Given $\mathbf{x}^* \in \mathcal{I}$, the value $f(\mathbf{x}^*)$ in Eq. (7) amounts to a simple function evaluation, realizing that $P(\mathbf{x}^*, t) = 0$ for $t \geq 3$.

**Step 1: Proving minimizers exist in $\mathcal{S}$.**

Let $\mathbf{x} \in \mathcal{D} \setminus \mathcal{S}$, and let $y \in [m]$ be an entry such that $x_y < 2/3$. Set $\delta = \min\{1 - \sum_{i=1}^m x_i, 2/3 - x_y\} > 0$ and define a new vector $\mathbf{x}' \in \mathcal{D}$ by

$$x_i' = \mathbf{x}'(i) = \begin{cases} x_i & \text{if } i \neq y \\ x_i + \delta & \text{otherwise.} \end{cases} \tag{8}$$

By our choice of $\delta$ we know that $\mathbf{x}' \in \mathcal{D}$, and we will prove this satisfies $f(\mathbf{x}') \leq f(\mathbf{x})$. To do so, we re-write $f(\mathbf{x})$ in a way that isolates terms involving $x_y$. Let $M_y = [m] \setminus \{y\}$. Recall that $f$ is a linear combination of terms $P(\mathbf{x}, t)$, where $P(\mathbf{x}, t)$ represents the probability that exactly $t$ events out of a set of $m$ independent events occur. Entry $x_i$ of $\mathbf{x}$ is the probability that $i$th event occurs. We can therefore re-write:

$$P(\mathbf{x}, t) = P(\mathbf{x}[M_y], t)(1 - x_y) + P(\mathbf{x}[M_y], t - 1)x_y. \tag{9}$$

To explain this in more detail, we use a slight abuse of terminology and refer to $[m]$ as a set of events. The first term in Eq. (9) is the probability that exactly $t$ events in $M_y$ occur and that event $y$ does not occur. The second term is the probability that $y$ does occur and exactly $t - 1$ events in $M_y$ occur. We therefore re-write $f(\mathbf{x})$ as

$$f(\mathbf{x}) = \sum_{t=0}^{m} a_t P(\mathbf{x}, t) \qquad \text{(definition of } f) \tag{10}$$

$$= \sum_{t=0}^{m} a_t \left( P(\mathbf{x}[M_y], t)(1 - x_y) + P(\mathbf{x}[M_y], t - 1)x_y \right) \qquad \text{(from Eq. (9))} \tag{11}$$

$$= \sum_{t=0}^{m} a_t P(\mathbf{x}[M_y], t)(1 - x_y) + \sum_{t=0}^{m} a_t P(\mathbf{x}[M_y], t - 1)x_y \qquad \text{(separating terms)} \tag{12}$$

$$= \sum_{t=0}^{m} a_t P(\mathbf{x}[M_y], t)(1 - x_y) + \sum_{t=-1}^{m-1} a_{t+1} P(\mathbf{x}[M_y], t)x_y \qquad \text{(change of variables)} \tag{13}$$

$$= \sum_{t=0}^{m-1} a_t P(\mathbf{x}[M_y], t)(1 - x_y) + \sum_{t=0}^{m-1} a_{t+1} P(\mathbf{x}[M_y], t)x_y \qquad (P(\mathbf{x}[M_y], m) = P(\mathbf{x}[M_y], -1) = 0) \tag{14}$$

$$= \sum_{t=0}^{m-1} P(\mathbf{x}[M_y], t) \left( a_t(1 - x_y) + a_{t+1} x_y \right) \qquad \text{(recombining)} \tag{15}$$

Observe that if we replace the value of $x_y$ with $x_y + \delta$ in Eq. (15) this gives the value of $f(\mathbf{x}')$. We can then see that

$$f(\mathbf{x}') = \sum_{t=0}^{m-1} P(\mathbf{x}[M_y], t) \left( a_t(1 - x_y - \delta) + a_{t+1}(x_y + \delta) \right)$$

$$= \sum_{t=0}^{m-1} P(\mathbf{x}[M_y], t) \left( a_t(1 - x_y) + a_{t+1} x_y + \delta(a_{t+1} - a_t) \right)$$

$$= f(\mathbf{x}) + \delta \sum_{t=0}^{m-1} (a_{t+1} - a_t).$$

Using the constraint $a_{t+1} \leq a_t$ we notice that the last term is always less that or equal to zero. Hence we have $f(\mathbf{x}') \leq f(\mathbf{x})$. Recall that $\delta = \min\{1 - \sum_{i=1}^m x_i, 2/3 - x_y\}$. If $\delta = 1 - \sum_{i=1}^m x_i$, then we can see that $\mathbf{x}' \in \mathcal{S}$ and we are done. Otherwise, $\delta = 2/3 - x_y$, and $x_y' = 2/3$. We can then apply the same exact procedure again, by increasing the value of some index $x_z$ (where $z \neq y$), which we know satisfies $x_z \leq 1/3$. The second application of this procedure is guaranteed to produce a vector in $\mathcal{S}$.

**Step 2: Editing a vector from $\mathcal{S}$ to $\mathcal{I}$.** The proof structure for Step 2 is very similar as Step 1, but is more involved as it requires working with two entries of $\mathbf{x}$ rather than one. Let $\mathbf{x} \in \mathcal{S} \setminus \mathcal{I}$, which means we can identify two entries $x_y$ and $x_z$ in the vector (with $y \neq z$) satisfying the inequality

$$0 < x_y \leq x_z < 2/3. \tag{16}$$

Given these entries, set $\delta = \min\{x_y, 2/3 - x_z\}$ and define a new vector $\mathbf{x}'$ by

$$x_i' = \mathbf{x}'(i) = \begin{cases} x_i & \text{if } i \notin \{y, z\} \\ x_i - \delta & \text{if } i = y \\ x_i + \delta & \text{if } i = z. \end{cases} \tag{17}$$

Observe that $\mathbf{x}' \in \mathcal{S}$ and that either $x_y' = 0$ or $x_z' = 2/3$. We will show that $f(\mathbf{x}') \leq f(\mathbf{x})$, by re-writing $f(\mathbf{x})$ in a way that isolates terms involving $x_y$ and $x_z$. Let $M_{yz} = [m] \setminus \{y, z\}$. Similar to the proof of Step 1, we can re-write $P(\mathbf{x}, t)$ as

$$P(\mathbf{x}, t) = P(\mathbf{x}[M_{yz}], t)(1 - x_y)(1 - x_z) \tag{18}$$
$$+ P(\mathbf{x}[M_{yz}], t - 1)\left[x_y(1 - x_z) + (1 - x_y)x_z\right] \tag{19}$$
$$+ P(\mathbf{x}[M_{yz}], t - 2)x_y x_z. \tag{20}$$

Line (18) captures the probability that exactly $t$ events from the set $M_{yz}$ happen, and that neither of the events $y$ or $z$ happen. Line (19) is the probability that $t - 1$ events from $M_{yz}$ happen, and exactly one of the events $\{y, z\}$ happens. Finally, line (20) is the probability that both $y$ and $z$ happen, and $t - 2$ of the events in $M_{yz}$ happen. For simplicity we now define

$$\alpha = (1 - x_y)(1 - x_z)$$
$$\beta = x_y(1 - x_z) + (1 - x_y)x_z$$
$$\gamma = x_y x_z,$$

so that we can re-write $f(\mathbf{x})$ as follows:

$$f(\mathbf{x}) = \sum_{t=0}^{m} a_t P(\mathbf{x}, t) \qquad\qquad \text{(definition of } f) \quad (21)$$

$$= \sum_{t=0}^{m} a_t \left(P(\mathbf{x}[M_{yz}], t)\alpha + P(\mathbf{x}[M_{yz}], t - 1)\beta + P(\mathbf{x}[M_{yz}], t - 2)\gamma\right) \qquad \text{(expanding } P(\mathbf{x}, t)) \quad (22)$$

$$= \sum_{t=0}^{m} a_t P(\mathbf{x}[M_{yz}], t)\alpha + \sum_{t=0}^{m} a_t P(\mathbf{x}[M_{yz}], t - 1)\beta + \sum_{t=0}^{m} a_t P(\mathbf{x}[M_{yz}], t - 2)\gamma \qquad \text{(separating three terms)} \quad (23)$$

$$= \sum_{i=0}^{m} a_i P(\mathbf{x}[M_{yz}], i)\alpha + \sum_{j=-1}^{m-1} a_{j+1} P(\mathbf{x}[M_{yz}], j)\beta + \sum_{k=-2}^{m-2} a_{k+2} P(\mathbf{x}[M_{yz}], k)\gamma \qquad \text{(change of variables)} \quad (24)$$

$$= \sum_{i=0}^{m} a_i P(\mathbf{x}[M_{yz}], i)\alpha + \sum_{j=0}^{m-1} a_{j+1} P(\mathbf{x}[M_{yz}], j)\beta + \sum_{k=0}^{m-2} a_{k+2} P(\mathbf{x}[M_{yz}], k)\gamma \tag{25}$$

$$= \sum_{i=0}^{m-2} a_i P(\mathbf{x}[M_{yz}], i)\alpha + \sum_{j=0}^{m-2} a_{j+1} P(\mathbf{x}[M_{yz}], j)\beta + \sum_{k=0}^{m-2} a_{k+2} P(\mathbf{x}[M_{yz}], k)\gamma \tag{26}$$

$$= \sum_{t=0}^{m-2} P(\mathbf{x}[M_{yz}], t)\left(a_t \alpha + a_{t+1}\beta + a_{t+2}\gamma\right) \qquad\qquad \text{(recombining sums).} \quad (27)$$

In Eq. (25) and Eq. (26) we have used the fact that $P(\mathbf{x}[M_{yz}], \ell) = 0$ if $\ell < 0$ or $\ell > m - 2$. Consider now the vector $\mathbf{x}'$ obtained by perturbing entries $x_y$ and $x_z$, and define

$$
\begin{aligned}
\alpha' &= (1 - x_y')(1 - x_z') & &= (1 - (x_y - \delta))(1 - (x_z + \delta)) & &= \alpha + \delta(x_y - x_z) - \delta^2 \\
\beta' &= x_y'(1 - x_z') + (1 - x_y')x_z' & &= (x_y - \delta)(1 - (x_z + \delta)) + (1 - (x_y - \delta))(x_z + \delta) & &= \beta + 2\delta(x_z - x_y) + 2\delta^2 \\
\gamma' &= x_y' x_z' & &= (x_y - \delta)(x_z + \delta) & &= \gamma + \delta(x_y - x_z) - \delta^2.
\end{aligned}
$$

In the expression for $f(\mathbf{x})$ given in Eq. (27), the entries $x_y$ and $x_z$ appear only in the terms $\alpha$, $\beta$, and $\gamma$. Therefore, if we replace $\{\alpha, \beta, \gamma\}$ with $\{\alpha', \beta', \gamma'\}$, we obtain a similar expression for $f(\mathbf{x}')$. We can then see that

$$
\begin{aligned}
f(\mathbf{x}') - f(\mathbf{x}) &= \sum_{t=0}^{m-2} P(\mathbf{x}[M_{yz}], t) \left( a_t(\alpha' - \alpha) + a_{t+1}(\beta' - \beta) + a_{t+2}(\gamma' - \gamma) \right) \\
&= \sum_{t=0}^{m-2} P(\mathbf{x}[M_{yz}], t) \left( a_t(\delta(x_y - x_z) - \delta^2) + a_{t+1}(2\delta(x_z - x_y) + 2\delta^2) + a_{t+2}(\delta(x_y - x_z) - \delta^2) \right) \\
&= \sum_{t=0}^{m-2} P(\mathbf{x}[M_{yz}], t) \left( a_t(\delta(x_y - x_z) - \delta^2) - 2a_{t+1}(\delta(x_y - x_z) - \delta^2) + a_{t+2}(\delta(x_y - x_z) - \delta^2) \right) \\
&= \sum_{t=0}^{m-2} P(\mathbf{x}[M_{yz}], t) \left( (a_t - 2a_{t+1} + a_{t+2})(\delta(x_y - x_z) - \delta^2) \right) \le 0,
\end{aligned}
$$

where in the last line we have used the fact that $a_t - 2a_{t+1} + a_{t+2} \ge 0$ and $(x_y - x_z) - \delta < 0$.

If we start with an arbitrary vector $\mathbf{x} \in \mathcal{S}$, applying one iteration of the above procedure will ensure that *at least* one entry of $\mathbf{x}$ will move to one of the endpoints $\{0, 2/3\}$ of the interval $[0, 2/3]$. Therefore, applying this procedure $m$ (or fewer) times to an arbitrary vector $\mathbf{x} \in \mathcal{S}$ will produce a vector $\mathbf{x}^* \in \mathcal{I}$ that satisfies $f(\mathbf{x}^*) \le f(\mathbf{x})$. $\qquad\square$

**Lemma B.6.** *The inequalities $a_t \ge a_{t+1}$ and $a_t + a_{t+2} \ge 2a_{t+1}$ hold for sequence $a_t = \frac{1}{1+g+t}$ where $g \ge 0$ is an arbitrary fixed integer. These inequalities also hold for the sequence*

$$
a_t = \frac{2}{9} \left( \frac{1}{t+1} + \frac{1}{t+3} \right) + \frac{5}{9} \left( \frac{1}{t+2} \right). \tag{28}
$$

*Proof.* The first inequality (monotonicity) is easy to check for both sequences. The most involved step is checking that the second inequality holds for the sequence in (28). To show it holds, we note that the inequality $a_t + a_{t+2} \ge 2a_{t+1}$ effectively corresponds to a discrete version of convexity. We first extend the definition of the sequence in (28) to all positive reals by defining the function

$$
h(x) = \frac{2}{9} \left( \frac{1}{x+1} + \frac{1}{x+3} \right) + \frac{5}{9} \left( \frac{1}{x+2} \right),
$$

which we can easily prove is convex over the interval $[0, \infty)$ by noting that its second derivative is

$$
h''(x) = \frac{1}{9} \left( \frac{10}{(x+2)^3} + \frac{4}{(x+3)^3} + \frac{4}{(x+1)^3} \right),
$$

which is greater than 0 for all $x \ge 0$. From the definition of convexity for any $\beta \in [0, 1]$ and any $x, y \ge 0$ we have

$$
h(x\beta + (1-\beta)y) \le \beta h(x) + (1-\beta)h(y).
$$

Now if we consider the specific values $x = t$, $y = t+2$, and $\beta = 1/2$ we get

$$
\begin{aligned}
h\left( t\frac{1}{2} + \frac{1}{2}(t+2) \right) &\le \frac{1}{2}h(t) + \frac{1}{2}h(t+2) \\
\implies h(t+1) &\le \frac{1}{2}h(t) + \frac{1}{2}h(t+2) \\
\implies 2h(t+1) &\le h(t) + h(t+2) \\
\implies 2a_{t+1} &\le a_t + a_{t+2}.
\end{aligned}
$$

One can use a similar approach to show (even more easily) that the sequence $a_t = 1/(1+g+t)$ also satisfies the second inequality. $\qquad\square$

**Proof of main approximation guarantee.** Given the above supporting lemmas, we are now ready to prove the approximation guarantee for graph MAXECC satisfied by Algorithm 2.

**Theorem 2.2.** For every $e \in E$, $\mathbb{P}[e \text{ is satisfied}] \geq pz_e$ where $p = 154/405 > 0.3802$ when running Algorithm 2.

*Proof of Theorem 2.2.* Fix $e = (u,v) \in E_c$ and set $C = [k] \setminus \{c\}$. As before, $X_v^i$ is the event that $v$ wants color $i$. Let $Y_v^i$ be the event that $v$ is *assigned* color $i$ by Algorithm 2, and $N_v$ be the event that $v$ wants *no* colors in $C$. If $c$ is strong for $v$, event $N_v$ is equivalent to event $Y_v^c$. Define $X_u^i$, $Y_u^i$, and $N_u$ analogously for $u$.

We break the proof up into three cases, which depend on whether 2, 1, or neither of the nodes in $e = (u,v)$ have $c = \ell(e)$ as a strong color.

**Case 1:** $c$ is strong for both $u$ and $v$, i.e., $S_u = S_v = \{c\}$.
Let $W_v'$ denote colors in $W_v$ that want $v$, and define $W_u'$ analogously for $u$. Since $c$ is the strong color for both $u$ and $v$, $e$ is satisfied if and only if $W_u' = W_v' = \emptyset$. Additionally, because nodes can only have a single strong color and we know $c$ is strong for both $u$ and $v$ we know that all other colors must be weak for both $u$ and $v$. Using the property that probabilities regarding separate colors are independent we have

$$\mathbb{P}[e \text{ is satisfied}] = \mathbb{P}[W_u' = W_v' = \emptyset] = \prod_{i \in C} \mathbb{P}[i \notin W_u' \cup W_v'].$$

Consider a color $i \in C_v$, which by definition means $x_v^i \geq x_u^i$. There are three options for the random threshold $\alpha_i$:

- $\alpha_i < x_u^i \leq x_v^i$, which happens with probability $x_u^i$ and implies that $i \in W_u' \cap W_v'$,

- $x_u^i < \alpha_i \leq x_v^i$, which happens with probability $x_v^i - x_u^i$ and implies $i \in W_v'$, $i \notin W_u'$, and

- $x_u^i \leq x_v^i < \alpha_i$, which happens with probability $1 - x_v^i$ and implies $i \notin W_u' \cup W_v'$.

Thus, for $i \in C_v$ we have $\mathbb{P}[i \notin W_u' \cup W_v'] = (1 - x_v^i)$ and similarly for $i \in C_u$ we have $\mathbb{P}[i \notin W_u' \cup W_v'] = (1 - x_u^i)$. This gives

$$\prod_{i \in C} \mathbb{P}[i \notin W_u' \cup W_v'] = \prod_{i \in C_v} (1 - x_v^i) \prod_{i \in C_u} (1 - x_u^i).$$

The fact that $c$ is strong for both $u$ and $v$ means $x_v^i \geq 2/3$ and $x_u^i \geq 2/3$. From the equality constraint in the LP we see that for $w \in \{u,v\}$ we have $\sum_{i \in C} x_w^i \leq 1 - 2/3 = 1/3$. Applying Lemma B.3 gives

$$\prod_{i \in C_v} (1 - x_v^i) \prod_{i \in C_u} (1 - x_u^i) \geq \frac{2}{3} \frac{2}{3} = \frac{4}{9} \geq \frac{4}{9} z_e.$$

**Case 2:** $c$ is strong for one of $u$ or $v$.
Without loss of generality we say $S_v = \{c\}$ and $c \in W_u$. Edge $e$ is satisfied if and only if $Y_u^c \cap Y_v^c$ holds. Because $c$ is strong for $v$, event $Y_v^c$ holds if and only if $N_v$ holds. Using the fact that $Y_u^c = Y_u^c \cap X_u^c$, we can write

$$\mathbb{P}[e \text{ is satisfied}] = \mathbb{P}[Y_u^c \cap X_u^c \cap N_v].$$

Using Bayes' Theorem, the fact that $X_u^c$ and $N_v$ are independent[2], Observation 2 ($\mathbb{P}[X_u^c] = x_u^c$), and Lemma B.2, we see that

$$
\begin{aligned}
\mathbb{P}[Y_u^c \cap X_u^c \cap N_v] &= \mathbb{P}[Y_u^c \mid N_v \cap X_u^c] \, \mathbb{P}[N_v \cap X_u^c] && \text{(Bayes' Theorem)} \\
&= \mathbb{P}[Y_u^c \mid N_v \cap X_u^c] \, \mathbb{P}[N_v] \, \mathbb{P}[X_u^c] && (X_u^c \text{ and } N_v \text{ are indepdendent}) \\
&= \mathbb{P}[Y_u^c \mid N_v \cap X_u^c] \, \mathbb{P}[N_v] \, x_u^c && \text{(Observation 1)} \\
&\geq \mathbb{P}[Y_u^c \mid N_v \cap X_u^c] \, \mathbb{P}[N_v] \, z_e && \text{(LP constraint } x_u^c \geq z_e) \\
&\geq \mathbb{P}[Y_u^c \mid X_u^c] \, \mathbb{P}[N_v] \, z_e && \text{(Lemma B.2).}
\end{aligned}
$$

---

[2] Independence follows from the fact that $X_u^c$ is concerned with color $c$, while $N_v$ is concerned with a disjoint color set $C = [k] \setminus \{c\}$.

Using $\overline{X}_v^i$ to indicate that $v$ *does not* want $i$, we get

$$\mathbb{P}\left[N_v\right] = \mathbb{P}\left[\bigcap_{i\in C}\overline{X}_v^i\right] = \prod_{i\in C}(1-x_v^i).$$

As in Case 1, because $c$ is strong for $v$, we know that $\sum_{i\in C}x_v^i \leq 1/3$ and applying Lemma B.3 gives us that $\mathbb{P}\left[N_v\right] \geq 2/3$.

Finally we must bound $\mathbb{P}\left[Y_u^c \mid X_u^c\right]$. Since $c$ is weak for $u$ (i.e., $c\in W_u$), it will be convenient to consider all weak colors for $u$ other than $c$, which we will denote by $\hat{W}_u = W_u\backslash\{c\}$. We will then use $\hat{W}_u' = \{i\in \hat{W}_u : \alpha_i < x_u^i\}$ to denote the set of colors in $\hat{W}_u$ that $u$ wants. Note that $X_u^c$ is independent of the number of colors in $\hat{W}_u$ that $u$ wants.

Because of the global ordering of colors, conditioned on $u$ wanting $c$ and wanting exactly $t$ colors in $\hat{W}_u$, there is a $1/(t+1)$ chance that $c$ is chosen first by the permutation $\pi$, and is therefore assigned to $u$. Formally:

$$\mathbb{P}\left[Y_u^c \mid X_u^c \cap (|\hat{W}_u'| = t)\right] = \frac{1}{t+1}.$$

Define $p_t = \mathbb{P}[|\hat{W}_u'| = t]$ to be the probability that exactly $t$ colors from $\hat{W}_u$ are wanted by $u$. Using the vector $\mathbf{x}_u[\hat{W}_u]$ to encode LP variables $\{x_u^i : i\in \hat{W}_u\}$ for node $u$ and the colors in $\hat{W}_u$, we see that $p_t = P(\mathbf{x}_v[\hat{W}_u], t)$. Therefore, the law of total probability (combined with the fact that $X_u^c$ is independent from $|\hat{W}_u'|$) gives

$$\mathbb{P}\left[Y_u^c \mid X_u^c\right] = \sum_{t=0}^{|\hat{W}_u|}\mathbb{P}\left[Y_u^c \mid X_u^c \cap (|\hat{W}_u'| = t)\right]\mathbb{P}[|\hat{W}_u'| = t] = \sum_{t=0}^{|\hat{W}_u|}\frac{1}{1+t}p_t. \tag{29}$$

Eq. (29) is exactly of the form in Lemma B.5 with $a_t = 1/(t+1)$. From Lemma B.6, we know this sequence $\{a_t\}$ satisfies the constraints needed by Lemma B.5, and applying Lemma B.5 shows $\mathbb{P}\left[Y_u^c \mid X_u^c\right] \geq 31/54$. Combining this with the previously established bound $\mathbb{P}\left[N_v\right] \geq 2/3$ gives

$$\mathbb{P}\left[e \text{ is satisfied}\right] \geq \mathbb{P}\left[Y_u^c \mid X_u^c\right]\mathbb{P}\left[N_v\right]z_e \geq \frac{31}{54}\cdot\frac{2}{3}z_e = \frac{31}{81}z_e > \frac{154}{405}z_e.$$

**Case 3:** $c$ is weak for both $u$ and $v$, i.e., $c\in W_u\cap W_v$.
First we condition the probability of satisfying $e$ on the event $X_u^c\cap X_v^c$. If one of $X_u^c$ or $X_v^c$ does not happen, then $e$ cannot be satisfied, so we have

$$\mathbb{P}\left[e \text{ is satisfied}\right] = \mathbb{P}\left[e \text{ is satisfied} \mid X_u^c\cap X_v^c\right]\mathbb{P}\left[X_v^c\cap X_u^c\right] = \mathbb{P}\left[e \text{ is satisfied} \mid X_u^c\cap X_v^c\right]z_e.$$

Define $D = (W_u\cup W_v)\backslash\{c\}$ to be the set of weak colors (excluding $c$) that want at least one of $\{u,v\}$. If we condition on $u$ and $v$ both wanting $c$, it is still possible that $e$ will not be satisfied. This will happen if a weak color $i\in D$ wants either $u$ or $v$, and the permutation $\pi$ prioritizes color $i$ over $c$. We bound the probability of satisfying $e$ by conditioning on the number of colors in $D$ that want one or both of $\{u,v\}$.

Using the same logic as in Observation 4, we can present a simpler way to characterize whether a color $i\in D$ wants at least one of $\{u,v\}$. Formally, we partition $D$ into the sets:

$$D_u = \{i\in D : x_u^i \geq x_v^i\}$$
$$D_v = D - D_v.$$

Note that a color $i\in D_u$ wants one or both nodes in $e$ if and only if $i$ wants $u$, and color $i\in D_v$ wants one or both nodes in $e$ if and only if $i$ wants $v$. Mirroring our previous notation, let $D_u'$ denote the set of colors in $D_u$ that want node $u$ (based on the random color thresholds) and define $D_v'$ analogously. The set $D_u'\cup D_v'$ is then the set of weak colors that want one or more node in $e$. Thus, conditioned on $|D_u'\cup D_v'| = t$ and conditioned on $u$ and $v$ wanting $c$, in order for $e$ to be satisfied $c$ must come before all $t$ colors in $|D_u'\cup D_v'|$ in the random permutation $\pi$. Formally, this means

$$\mathbb{P}\left[e \text{ is satisfied} \mid X_u^c\cap X_v^c \cap (|D_u'\cup D_v'| = t)\right] = \frac{1}{1+t}.$$

Again using the law of total probability and the fact that $X_u^c \cap X_v^c$ is independent from $|D_u' \cup D_v'|$, we see

$$\mathbb{P}[e \text{ is satisfied} \mid X_u^c \cap X_v^c] = \sum_{t=0}^{|D|} \frac{1}{1+t} \mathbb{P}\left[|D_u' \cup D_v'| = t\right]. \tag{30}$$

Observe now that $D_u$ and $D_v$ are disjoint color sets, which implies that $|D_u'|$ and $|D_v'|$ are independent random variables. This allows us to decouple $D_u'$ and $D_v'$ in the above expression since $\mathbb{P}\left[|D_u' \cup D_v'| = t\right] = \mathbb{P}\left[|D_u'| + |D_v'| = t\right]$. We define

$$p_i = \mathbb{P}\left[|D_u'| = i\right] \text{ and } q_i = \mathbb{P}\left[|D_v'| = i\right],$$

so that we can write

$$\mathbb{P}\left[|D_u'| + |D_v'| = t\right] = \sum_{\substack{i,j \in [0,t] \\ i+j=t}} p_i q_j = \sum_{i=0}^{|D_u|} p_i q_{t-i}.$$

This allows us to re-write Eq. (30) as

$$\mathbb{P}[e \text{ is satisfied} \mid X_u^c \cap X_v^c] = \sum_{i=0}^{|D_u|} \sum_{j=0}^{|D_v|} \frac{1}{1+i+j} p_i q_j. \tag{31}$$

Without loss of generality we now assume $|D_u| \leq |D_v|$, and proceed case by case depending on the sizes of $D_u$ and $D_v$. Our goal is to show that for all cases $\mathbb{P}[e \text{ is satisfied} \mid X_u^c \cap X_v^c] \geq 154/405$. In the cases below, we use the vector $\mathbf{x}_u[D_u]$ to encode LP variables for node $u$ and colors in $D_u$ so that $p_i = P(\mathbf{x}_u[D_u], i)$, and likewise $q_i = P(\mathbf{x}_v[D_v], i)$, to match with notation in Definition B.4 and Lemma B.5.

**Case 3a.** $|D_u| = |D_v| = 0$.
No weak colors other than $c$ want $u$ or $v$, so $\mathbb{P}[e \text{ is satisfied} \mid X_u^c \cap X_v^c] = 1$.

**Case 3b.** $|D_u| = 0$ and $|D_v| = 1$.
Letting $a$ be the single color in $D_v$,

$$\mathbb{P}[e \text{ is satisfied} \mid X_u^c \cap X_v^c] = p_0 q_0 + \frac{1}{2} p_0 q_1 = q_0 + \frac{1}{2} q_1 = (1 - x_v^a) + \frac{1}{2} x_v^a.$$

Because $a$ is a weak color for $v$, we know $x_v^a \in [0, \frac{2}{3}]$, and the minimum value the probability can obtain is $\frac{2}{3} > \frac{154}{405}$.

**Case 3c.** $|D_u| = |D_v| = 1$.
Letting $a$ be the color in $D_v$ and $b$ be the color in $D_u$,

$$\mathbb{P}[e \text{ is satisfied} \mid X_u^c \cap X_v^c] = p_0 q_0 + \frac{1}{2}(p_0 q_1 + p_1 q_0) + \frac{1}{3} p_1 q_1$$

$$= (1 - x_u^b)(1 - x_v^a) + \frac{1}{2}((1 - x_u^b) x_v^a + x_u^b(1 - x_v^a)) + \frac{1}{3} x_u^b x_v^a.$$

Because $a$ is weak for $v$ and $b$ is weak for $u$, we know $x_v^a, x_u^b \in [0, \frac{2}{3}]$. This gives a minimum value for the probability of $\frac{13}{27} > \frac{154}{405}$.

**Case 3d.** $|D_u| = 0$ and $|D_v| > 1$.
In this case $p_0 = 1$, and Eq. (31) simplifies to

$$\mathbb{P}[e \text{ is satisfied} \mid X_u^c \cap X_v^c] = \sum_{j=0}^{|D_v|} \frac{1}{j+1} q_j \geq \frac{31}{54},$$

where we have applied Lemma B.5 with $a_j = \frac{1}{j+1}$.

**Case 3e.** $|D_u| = 1$ and $|D_v| > 1$.
Eq. (31) simplifies to

$$\mathbb{P}[e \text{ is satisfied} \mid X_u^c \cap X_v^c] = p_0 \sum_{j=0}^{|D_v|} \frac{q_j}{j+1} + p_1 \sum_{j=0}^{|D_v|} \frac{q_j}{j+2} \geq p_0 \frac{31}{54} + p_1 \frac{19}{54},$$

where we applied Lemma B.5 twice, once with $a_j = \frac{1}{j+1}$ and once with $a_j = \frac{1}{j+2}$. The right hand side of the above bound has a minium value of $\frac{23}{54}$.

**Case 3f.** $|D_u| > 1$ and $|D_v| > 1$. From Eq. (31) we know

$$\mathbb{P}[e \text{ is satisfied} \mid X_u^c \cap X_v^c] = \sum_{i=0}^{|D_u|} p_i \left( \sum_{j=0}^{|D_v|} \frac{1}{1+i+j} q_j \right).$$

We then apply Lemma B.5 $|D_u| + 1$ times, once for each choice of $i \in \{0, 1, \ldots, |D_u|\}$. The $i$th time we apply it, we use the sequence $a_j = \frac{1}{1+i+j}$ to get

$$\sum_{j=0}^{|D_v|} \frac{1}{1+i+j} q_j \geq \frac{2}{9} \left( \frac{1}{i+1} + \frac{1}{i+3} \right) + \frac{5}{9} \left( \frac{1}{i+2} \right). \tag{32}$$

The right hand of (32) defines a new sequence

$$a_i = \left( \frac{1}{i+1} + \frac{1}{i+3} \right) + \frac{5}{9} \left( \frac{1}{i+2} \right). \tag{33}$$

We know from Lemma B.6 that this sequence $\{a_i\}$ in Eq. (33) satisfies the conditions from Lemma B.5, so applying Lemma B.5 one more time and combining all steps gives

$$\mathbb{P}[e \text{ is satisfied} \mid X_u^c \cap X_v^c] = \sum_{i=0}^{|D_u|} p_i \left( \sum_{j=0}^{|D_v|} \frac{1}{1+i+j} q_j \right) \geq \sum_{i=0}^{|D_u|} p_i \left( \frac{2}{9} \left( \frac{1}{i+1} + \frac{1}{i+3} \right) + \frac{5}{9} \left( \frac{1}{i+2} \right) \right) \geq \frac{154}{405}.$$

Therefore, we have shown in all possible subcases of Case 3 that $\mathbb{P}\left[e \text{ is satisfied} \mid X_u^c \cap X_v^c\right] \geq \frac{154}{405}$.

Cases 1,2, and 3 all together show that for every $e \in E$ we have

$$\mathbb{P}\left[e \text{ is satisfied}\right] \geq \frac{154}{405} z_e.$$

$\square$

## C. Omitted proofs from Section 3.1

**Theorem 3.1.** $\ell_p$-NORM MINECC *is approximable within factor* 2 *when* $p \geq 1$, *and factor* $2^{1/p}$ *when* $0 < p < 1$.

*Proof.* We begin with the case where $p \geq 1$. It is well known that the $\ell_p$-norm is convex, and we observe that all of the constraints in Program (3) are linear. Thus, we may in polynomial time obtain optimal fractional variables $\{d_v^c\}, \{\gamma_e\}, \{m_c\}$ for the relaxation of Program (3).

We observe that for each vertex $v$, there is at most one color $c$ with the property that $d_v^c < \frac{1}{2}$. Otherwise, there are two colors $c_1, c_2$ with $d_v^{c_1}, d_v^{c_2} < \frac{1}{2}$, but then

$$\sum_{c=1}^{k} d_v^c < 2 \cdot \frac{1}{2} + (k-2) = k-1,$$

contradicting the first constraint of Program (3).

We now propose a coloring $\lambda$ as follows. For each color $c$ and vertex $v$, we assign $\lambda(v) = c$ if $d_v^c < \frac{1}{2}$. By the preceding observation, this procedure assigns at most one color to each vertex. If for any $v$ we have that $d_v^c \geq \frac{1}{2}$ for all colors $c$, then we set $\lambda(v)$ arbitrarily.

Now, for each hyperedge $e$ let $\hat{\gamma}_e$ be equal to 0 if $\lambda(v) = \ell(e)$ for all $v \in e$, or 1 otherwise. We claim that if $\hat{\gamma}_e = 1$, then $\gamma_e \geq \frac{1}{2}$. Otherwise, letting $c = \ell(e)$, the second constraint of Program (3) implies that $d_v^c < \frac{1}{2}$ for all $v \in e$. Then $\lambda(v) = c$ for all $v \in e$, implying that $\hat{\gamma}_e = 0$, a contradiction. We conclude that for every hyperedge $e$, $\hat{\gamma}_e \leq 2\gamma_e$.

For each color $c$, we write $\hat{m}_c$ for the number of hyperedges of color $c$ which are unsatisfied by $\lambda$. Equivalently,

$$\hat{m}_c = \sum_{e \in E_c} \hat{\gamma}_e \leq 2 \sum_{e \in E_c} \gamma_e = 2m_c,$$

where the last equality comes from the third constraint of Program 3.

It follows that for every $c$, $\hat{m}_c^p \leq 2^p m_c^p$, and thus

$$\sum_{c=1}^{k} \hat{m}_c^p \leq 2^p \cdot \sum_{c=1}^{k} m_c^p,$$

which in turn implies that

$$\left( \sum_{c=1}^{k} \hat{m}_c^p \right)^{1/p} \leq 2 \cdot \left( \sum_{c=1}^{k} m_c^p \right)^{1/p},$$

so $\lambda$ is 2-approximate.

We now consider the case where $0 < p < 1$. We begin by defining, for each $c \in [k]$, a non-negative, monotone, and submodular function $f_c \colon 2^E \to \mathbb{Z}$ given by $f_c(S) = |E_c \cap S|$. If $\lambda$ is a vertex coloring of $H$ and $S_\lambda \subseteq E$ is the set of edges unsatisfied by $\lambda$, then $f_c(S_\lambda)$ measures the number of edges of color $c$ unsatisfied by $\lambda$. Because the composition of a concave function with a submodular function results in a submodular function and $0 < p < 1$, $f_c^p$ is non-negative, monotone, and submodular. Then the function $f \colon 2^E \to \mathbb{R}$ given by the sum

$$f(S) = \sum_{c=1}^{k} f_c^p(S)$$

has these same properties.

Now, let $\lambda^*$ be an optimal vertex coloring of $H$ and let $S_{\lambda^*}$ be the set of hyperedges unsatisfied by $\lambda^*$. Our goal will be to compute a coloring $\lambda$ with $f(S_\lambda) \leq 2f(S_{\lambda^*})$. This would complete the proof, since then we have

$$(f(S_\lambda))^{1/p} \leq (2f(S_{\lambda^*}))^{1/p} = 2^{1/p} f^{1/p}(S_{\lambda^*}),$$

and $f^{1/p}(S_\lambda)$, $f^{1/p}(S_{\lambda^*})$ are precisely the objective values attained by $\lambda$ and $\lambda^*$, respectively, according to Program (3).

We give a classic analysis based on the Lovász extension (Lovász, 1983). Some readers may find it helpful to observe that the following is conceptually equivalent to the textbook 2-approximation for SUBMODULAR VERTEX COVER[3], where the graph in question is the *conflict graph* $G$ of our edge-colored hypergraph $H$. The conflict graph $G$ is obtained by replacing each hyperedge in $H$ with a node in $G$, and adding an edge between two nodes in $G$ if they correspond to overlapping hyperedges of different colors in $H$. The connection between vertex colorings of edge-colored hypergraphs and vertex covers of their associated conflict graphs has been observed several times in the literature (Angel et al., 2016; Cai & Leung, 2018; Kellerhals et al., 2023; Veldt, 2023), and also appears elsewhere in the present work, e.g., in the proofs of Theorems 3.8, D.2 and D.3.

Impose an arbitrary order on the edges of $E$ and for each vector $\gamma \in [0,1]^{|E|}$ map edges to components of $\gamma$ according to this order. Henceforth, for a hyperedge $e$ we write $\gamma_e$ for the component of $\gamma$ corresponding to $e$. Moreover, for a value $\rho$ selected uniformly at random from $[0,1]$, we write $T_\rho(\gamma) = \{e \colon \gamma_e \geq \rho\}$. Then the Lovász extension $\hat{f} \colon [0,1]^{|E|} \to \mathbb{R}$ of $f$ is given by

$$\hat{f}(\gamma) = \mathbb{E}[f(T_\rho(\gamma))].$$

This $\hat{f}$ is convex (Lovász, 1983), so we may efficiently optimize the following program:

$$
\begin{aligned}
\min \quad & \hat{f}(\gamma) \\
\text{s.t.} \quad & \gamma_e + \gamma_f \geq 1 \quad \text{for all } e, f \in E \text{ such that } e \cap f \neq \emptyset \text{ and } \ell(e) \neq \ell(f) \\
& \gamma_e \geq 0 \qquad \text{for all } e \in E.
\end{aligned}
\tag{34}
$$

---

[3]Given a graph $G$ and a submodular function $f$, find a vertex cover $C$ minimizing $f(C)$.

Consider the binary vector given by $\gamma_e = 1$ if $e \in S_{\lambda_*}$ or 0 otherwise; note that this vector satisfies the constraints of Program (34). Hence, we can see that the minimum value for Program (34) provides a lower bound on $f(S_{\lambda_*})$.

Let $\gamma$ be a minimizer of Program (34), and let

$$S = \left\{ e \in E \colon \gamma_e \geq \frac{1}{2} \right\}.$$

The first constraint of Program (34) ensures that $S$ contains at least one member of every pair of overlapping and distinctly colored hyperedges. Thus, it is trivial to compute a coloring $\lambda$ with $S_\lambda \subseteq S$. It follows from monotonicity that $f(S_\lambda) \leq f(S)$, and that for each $\rho \leq \frac{1}{2}$, $f(S) \leq f(T_\rho(\gamma))$. Putting it all together, we have

$$\frac{f(S_\lambda)}{2} \leq \frac{f(S)}{2} = \int_0^{\frac{1}{2}} f(S) \, \mathrm{d}\rho \leq \int_0^{\frac{1}{2}} f(T_\rho(\gamma)) \, \mathrm{d}\rho \leq \int_0^1 f(T_\rho(\gamma)) \, \mathrm{d}\rho = \hat{f}(\gamma) \leq f(S_{\lambda_*}),$$

which completes the proof.

$\square$

**Theorem 3.2.** *As $p \to \infty$, the integrality gap of Program (3) converges to* 2.

*Proof.* Consider a triangle on three vertices $v_1, v_2, v_3$ with distinct edge colors $c_{12}, c_{23}, c_{13}$. It is not possible to satisfy more than one edge, so the optimal objective value is $(1^p + 1^p + 0^p)^{1/p} = 2^{1/p}$. Meanwhile, a fractional relaxation of Program (3) can set $d_{v_1}^{c_{12}} = d_{v_1}^{c_{13}} = d_{v_2}^{c_{12}} = d_{v_2}^{c_{23}} = d_{v_3}^{c_{13}} = d_{v_3}^{c_{23}} = \frac{1}{2}$, thus achieving (fractional) objective value $(\frac{1}{2^p} + \frac{1}{2^p} + \frac{1}{2^p})^{1/p} = \frac{3^{1/p}}{2}$. Thus, the ratio of the optimal integral objective to the optimal fractional objective is $2 \cdot (\frac{2}{3})^{1/p}$, which converges to 2 as $p$ approaches $\infty$. $\square$

# D. Omitted proofs from Section 3.2

**Theorem 3.3.** COLOR-FAIR MINECC *is* NP-*hard even when restricted to subcubic trees with cutwidth* 2, *exactly* 3 *edges of each color, and* $\tau = 2$. COLOR-FAIR MAXECC *is* NP-*hard even when restricted to paths with* $\tau = 1$.

*Proof.* We reduce from BOOLEAN SATISFIABILITY, for which the input is a formula written in conjunctive normal form, consisting of $m$ clauses $C_1, C_2, \ldots C_m$ over $n$ variables $x_1, x_2, \ldots x_n$. For each variable $x_i$, we write $x_i$ and $\neg x_i$ for the corresponding positive and negative literals. We make the standard assumptions that no clause contains both literals of any variable, and that every literal appears at least once. We also assume that each clause has size 2 or 3, and that each variable appears in at most three clauses (implying that each literal appears in at most two clauses); hardness is retained under these assumptions (Tovey, 1984). Given an instance of this problem, we construct an instance $(G = (V, E), \tau = 2)$ of COLOR-FAIR MINECC. See Figure 3 for a visual aid.

For each clause $C_j$, we create a unique color $c_j$. Also, if $C_j$ has size two, we create a second unique color $c'_j$, and five vertices $v_j^a, v_j^b, v_j^c, v_j^d$, and $v_j^e$. We call these the *spare vertices* associated with $C_j$. The reason for creating these vertices and the second color $c'_j$ will become apparent later. Next, for each variable $x_i$, we create a vertex $v_{j_1, j_2}^i$ for each (ordered) pair of clauses $C_{j_1}, C_{j_2}$ with $C_{j_1}$ containing the positive literal $x_i$ and $C_{j_2}$ containing the negative literal $\neg x_i$. We call $v_{j_1, j_2}^i$ a *conflict vertex* for the variable $x_i$ and the variable-clause pairs $(x_i, C_{j_1})$ and $(x_i, C_{j_2})$. Observe that each variable has either one or two associated conflict vertices, as does each variable-clause pair. For each variable-clause pair $(x_i, C_j)$ with a single associated conflict vertex, we create an additional vertex $v_j^i$ and call this the *free vertex* for the variable-clause pair $(x_i, C_j)$.

Now we create edges. For each clause $C_j$, we begin by creating one edge $e_j^i$ of color $c_j$ for each variable $x_i$ contained in $C_j$. This edge contains either the two $(x_i, C_j)$ conflict vertices, or the single $(x_i, C_j)$ conflict vertex and the $(x_i, C_j)$ free vertex. We call the edge $e_j^i$ a *conflict edge* associated both with clause $C_j$, and with the literal of $x_i$ which appears in $C_j$. Observe that every free vertex has degree one, and every conflict vertex has degree two. For clauses $C_j$ of size 3, there are exactly three edges of color $c_j$. For clauses $C_j$ of size 2, we have thus far created two edges of color $c_j$. For each such clause, we use the associated spare vertices to create a third edge $\{v_j^a, v_j^b\}$ of color $c_j$, which we call the *spare edge* associated with $C_j$, and three edges $\{v_j^a, v_j^c\}$, $\{v_j^a, v_j^d\}$, and $\{v_j^b, v_j^e\}$, each of color $c'_j$. Observe that our constructed graph still has maximum degree three, and that there are now exactly 3 edges of every color. Finally, we set $\tau = 2$.

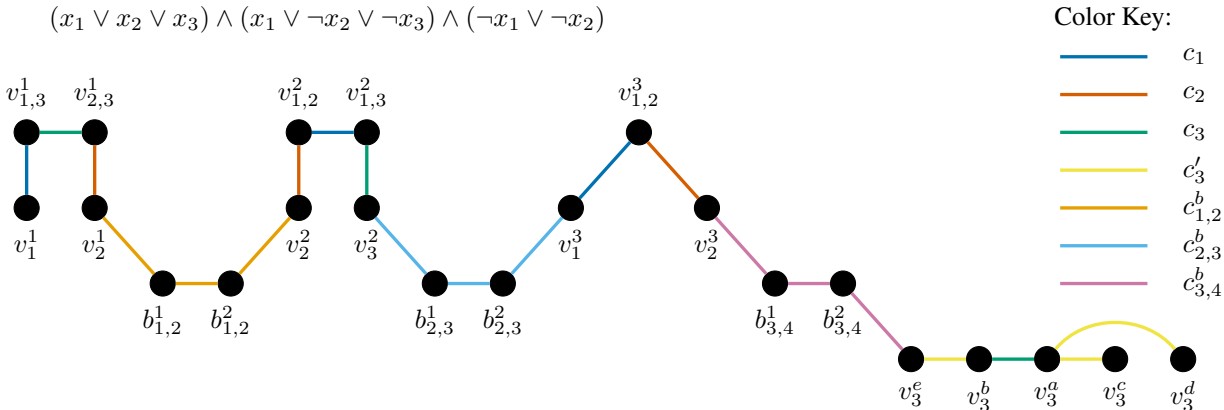

**Figure 3.** The construction given by Theorem 3.3 for the CNF formula on three clauses $C_1 = (x_1 \lor x_2 \lor x_3)$, $C_2 = (x_1 \lor \neg x_2 \lor \neg x_3)$, and $C_3 = (\neg x_1 \lor \neg x_2)$. Notice that the vertices are partitioned visually into four horizontal layers. The top layer contains *conflict* vertices, the second from top contains *free* vertices, the second from bottom *bridge* vertices, and the bottom *spare* vertices. Every edge containing a conflict vertex is a *conflict* edge, every edge containing a bridge vertex is a *bridge* edge, and all other edges are *spare* edges. The colors $c_1, c_2$ and $c_3$ correspond to the clauses $C_1, C_2$, and $C_3$. The color $c_3'$ is the *spare* color associated with $C_3$. The remaining colors are *bridge* colors. We refer to the proof of Theorem 3.3 for a formal description of the construction and the accompanying analysis.

Since free vertices have degree one, they do not participate in cycles. It is also clear from the construction that spare vertices do not participate in cycles. Thus, any cycle contains only conflict vertices. Note that if two conflict vertices share an edge, then they are associated with the same variable-clause pair. It follows that every vertex in a cycle is associated with some single variable-clause pair. However, a variable-clause pair has at most two associated conflict vertices, so we have constructed a forest.

Now we will add some gadgets to turn our forest into a tree. Suppose that the graph we have constructed so far has $q$ connected components. Impose an arbitrary order on these components, and label them $G_1, G_2, \ldots, G_q$. Observe that every connected component contains either only spare vertices or two free vertices. In both cases, it is possible to add two edges each with one (distinct) endpoint in $G_i$ without raising the maximum degree above three. That the endpoints are distinct will be important when we analyze the cutwidth. For each consecutive pair $G_i, G_{i+1}$ of connected components, we add two *bridge vertices* $b_{i,i+1}^1, b_{i,i+1}^2$ and a *bridge color* $c_{i,i+1}^b$. We add *bridge edges* (chosen so as not to violate our maximum degree constraint) from $G_i$ to $b_{i,i+1}^1$, from $b_{i,i+1}^1$ to $b_{i,i+1}^2$, and from $b_{i,i+1}^2$ to $G_{i+1}$. We color each of these three edges with $c_{i,i+1}^b$. Observe that the graph is now connected, but it is still acyclic since there is exactly one path between any pair of vertices which were in different connected components before the addition of our bridge gadgets.

We now claim that the constructed graph $G$ has cutwidth 2. To this end, we will construct a ordering $\sigma \colon V \to \mathbb{N}$ of the vertices of $G$, where $\sigma$ is injective, $\sigma(v) = 1$ indicates that $v$ is the first vertex in the ordering, $\sigma(u) = |V|$ indicates that $u$ is the last vertex in the ordering, and $\sigma(v) < \sigma(u)$ if and only if $v$ precedes $u$ in the ordering. We will show that for every $i$, there exist at most two edges $uv \in E$ with the property that $\sigma(u) \leq i$ and $\sigma(v) > i$. We refer to this number of edges as the *width* of the cut between vertices $i$ and $i + 1$ in $\sigma$. We begin by guaranteeing that, for every $i$, if $u$ is the last vertex of $G_i$ in $\sigma$, $v$ is the first vertex of $G_{i+1}$ in $\sigma$, and $b_{i,i+1}^1, b_{i,i+1}^2$ are the associated bridge vertices, then $\sigma(u) = \sigma(b_{i,i+1}^1) - 1 = \sigma(b_{i,i+1}^2) - 2 = \sigma(v) - 3$. A consequence is that for every $i$, if $u, v$ are vertices of $G_i$ then every vertex $w$ with the property that $\sigma(u) < \sigma(w) < \sigma(v)$ is also a vertex of $G_i$. Observe that every cut between two bridge vertices has width 1, as does every cut between a bridge vertex and a non-bridge vertex. We therefore need only consider cuts between pairs of vertices in the same $G_i$. If $G_i$ contains a conflict vertex, then $G_i$ is either a $P_3$ or a $P_4$. The former case arises when a variable appears in exactly two clauses, so there is one associated conflict vertex adjacent to two free vertices. The latter case arises when a variable appears in exactly three clauses, so there are two associated conflict vertices. These are adjacent. Additionally, each conflict vertex is adjacent to a distinct free vertex, so we have a $P_4$. Both the $P_3$ and $P_4$ have cutwidth 1, as evidenced by ordering the vertices as they appear along the path. Moreover, we may assume that the relevant bridge edges are incident on the appropriate endpoints of the path, and so any cut between vertices of $G_i$ has width 1. Otherwise, $G_i$ contains no conflict vertices. In this case, $G_i$ contains only the five spare vertices $v_j^a, v_j^b, v_j^c, v_j^d$, and $v_j^e$ associated with some clause $C_j$, and the four edges are $v_j^a v_j^b, v_j^a v_j^c, v_j^a v_j^d$, and $v_j^b v_j^e$. It is simple to check that this

construction has cutwidth 2, as evidenced by the ordering $\sigma(v_j^e) < \sigma(v_j^b) < \sigma(v_j^a) < \sigma(v_j^c) < \sigma(v_j^d)$. Once again, we may assume that the relevant bridge edges are incident on $v_j^e$ and $v_j^d$. Moreover, we can assume that at least one such $G_i$ exists, since BOOLEAN SATISFIABILITY instances in which every clause has size three and every variable appears at most three times are polynomial-time solvable (Tovey, 1984). Thus, $G$ has cutwidth 2.

It remains to show that the reduction is correct. For the first direction, assume that there exists an assignment $\phi$ of boolean values to the variables $x_1, x_2, \ldots, x_n$ which satisfies every clause. We say that the assignment $\phi$ *agrees* with a variable-clause pair $(x_i, C_j)$ if $C_j$ contains the positive literal $x_i$ and $\phi(x_i) = \mathsf{True}$ or if $C_j$ contains the negative literal $\neg x_i$ and $\phi(x_i) = \mathsf{False}$. We color the vertices of our constructed graph as follows. To every free vertex associated with clause $C_j$ we assign color $c_j$. The free vertex was only in edges of color $c_j$, so this results in zero unsatisfied edges. Next, to each bridge vertex we assign the associated bridge color. The bridge vertices were only in edges of this color, so once again this results in zero unsatisfied edges. Moreover, we have now guaranteed that at least one bridge edge of every bridge color is satisfied, meaning that at most two can be dissatisfied. Henceforth, we will not consider the remaining bridge edges. Next, for each clause $C_j$ of size 2, we assign color $c_j'$ to every spare vertex associated with $C_j$. This results in exactly one unsatisfied edge of color $c_j$, and ensures that every edge of color $c_j'$ is satisfied. Finally, for each conflict vertex $v_{j_1, j_2}^i$, if $\phi$ agrees with $(x_i, C_{j_1})$ we assign color $c_{j_1}$ to $v_{j_1, j_2}^i$, and otherwise we assign color $c_{j_2}$. Observe that this coloring satisfies an edge $e_j^i$ if and only if $\phi$ agrees with $(x_i, C_j)$. Moreover, because $\phi$ is satisfying, every clause $C_j$ contains at least one variable $x_i$ such that $\phi$ agrees with $(x_i, C_j)$. Hence, at least one edge of every color is satisfied. Because there are exactly three edges of every color, there are at most two unsatisfied edges of any color.

For the other direction, assume that we have a coloring which leaves at most two edges of any color unsatisfied. We will create a satisfying assignment $\phi$. For each variable $x_i$, we set $\phi(x_i) = \mathsf{True}$ if any conflict edge associated with the positive literal $x_i$ is satisfied, and $\phi(x_i) = \mathsf{False}$ otherwise. We now show that $\phi$ is satisfying. Consider any clause $C_j$. There are exactly three edges with color $c_j$, and at least one of them is satisfied. We may assume that the spare edge associated with $C_j$ (if such an edge exists) is unsatisfied, since satisfying this edge would require three unsatisfied edges of color $c_j'$. Thus, at least one conflict edge associated with $C_j$ is satisfied. Let $x_i$ be the corresponding variable, so the satisfied conflict edge is $e_j^i$. If $C_j$ contains the positive literal $x_i$, then $\phi(x_i) = \mathsf{True}$ so $C_j$ is satisfied by $\phi$. Otherwise $C_j$ contains the negative literal $\neg x_i$. In this case, we observe that every conflict edge associated with the positive literal $x_i$ intersects with $e_j^i$ at a conflict vertex, and none of these edges has color $c_j$ since $C_j$ does not contain both literals. Hence, the satisfaction of $e_j^i$ implies that every conflict edge associated with the positive literal $x_i$ is unsatisfied. It follows that $\phi(x_i) = \mathsf{False}$, meaning $C_j$ is satisfied by $\phi$.

To show the claim for COLOR-FAIR MAXECC, we repeat the same construction, except that we omit all spare colors, vertices, and edges. The effect is that the constructed graph is a path. The proof of correctness is conceptually unchanged. Given a satisfying assignment $\phi$ we color vertices in the same way as before, and given a vertex coloring which satisfies at least one edge of every color, it remains the case that at least one conflict edge associated with every clause must be satisfied. The remaining analysis is similar. $\qquad\square$

**Theorem 3.4.** COLOR-FAIR MINECC *is solvable in* $O(2^t r |E|)$*-time.*

*Proof.* We give a branching algorithm. Given an instance $(H = (V, E), \tau)$ of COLOR-FAIR MINECC, a *conflict* is a triple $(v, e_1, e_2)$ consisting of a single vertex $v$ and a pair of distinctly colored hyperedges $e_1, e_2$ which both contain $v$. If $H$ contains no conflicts, then it is possible to satisfy every edge. Otherwise, we identify a conflict in $O(r|E|)$ time by scanning the set of hyperedges incident on each node. Once a conflict $(v, e_1, e_2)$ has been found, we branch on the two possible ways to resolve this conflict: deleting $e_1$ or deleting $e_2$. Here, deleting a hyperedge has the same effect as "marking" it as unsatisfied and no longer considering it for the duration of the algorithm. We note that it is simple to check in constant time whether a possible branch violates the constraint given by $\tau$; these branches can be pruned. Because each branch increases the number of unsatisfied hyperedges by 1, the search tree has depth at most $t$. Thus, by computing the search tree in level-order, the algorithm runs in time $O(2^t r |E|)$. $\qquad\square$

---

**FAIR BIPARTITE VERTEX COVER**

**Input:** A bipartite graph $G = (V = A \uplus B, E)$ and an integer $\alpha$.

**Problem:** Does there exist a vertex cover $C$ of $G$ with $\max\{|C \cap A|, |C \cap B|\} \leq \alpha$?

---

**Theorem D.1.** FAIR BIPARTITE VERTEX COVER *is* NP*-hard.*

*Proof.* We reduce from CONSTRAINED BIPARTITE VERTEX COVER, for which the input is a bipartite graph $G = (A \uplus B, E)$ along with two integers $\alpha_a, \alpha_b$, and the question is whether there exists a vertex cover $C$ with $|C \cap A| \leq \alpha_a$ and $|C \cap B| \leq \alpha_b$. This problem was shown to be NP-complete by Kuo & Fuchs (1987).

Given an instance $(G = (A \uplus B, E), \alpha_a, \alpha_b)$ of CONSTRAINED BIPARTITE VERTEX COVER, we construct an instance $(G' = (A' \uplus B', E'), \alpha)$ of FAIR BIPARTITE VERTEX COVER as follows. First, we observe that if $\alpha_a = \alpha_b$ then $(G, \alpha_a, \alpha_b)$ is already an instance of FAIR BIPARTITE VERTEX COVER, in which case there is nothing to do. We therefore set $\beta = \alpha_b - \alpha_a$ and assume without loss of generality that $\beta > 0$. We begin by copying $G'$. That is, for each vertex $a \in A$ we create a vertex $a' \in A'$, for each vertex $b \in B$ we create a vertex $b' \in B'$, and for each edge $ab \in E$ we create an edge $a'b' \in E'$. We call the vertices (and edges) that we have created thus far *original* vertices (and edges). Next, we create $\beta$ auxiliary vertices $\{a_i \mid 1 \leq i \leq \beta\} \subset A'$, and $\beta(\alpha_b + 1)$ auxiliary vertices $\{b_{ij} \mid 1 \leq i \leq \beta, 1 \leq j \leq \alpha_b + 1\} \subset B'$. We also add auxiliary edges $\{a_i b_{ij} \mid 1 \leq i \leq \beta, 1 \leq j \leq \alpha_b + 1\} \subset E'$. Finally, we set $\alpha = \alpha_b$. This concludes the construction.

It remains to show that the reduction is correct. For the first direction, assume that $C \subseteq V$ is a vertex cover of $G$ with $|C \cap A| \leq \alpha_a$ and $|C \cap B| \leq \alpha_b$. We construct a vertex cover $C'$ of $G'$ as follows. For each vertex $a \in C \cap A$, we add $a'$ to $C$, and for each vertex $b \in C \cap B$, we add $b'$ to $C$. We also add the auxiliary vertex $a_i$ to $C$, for each $1 \leq i \leq \beta$. Because $C$ is a vertex cover of $E$, $C'$ covers every original edge in $E'$. Because every auxiliary edge in $E'$ is incident on some auxiliary vertex $a_i$, $C'$ also covers all auxiliary edges in $E'$. Hence, $C'$ is a vertex cover of $G'$. Moreover, by construction $|C' \cap B'| \leq \alpha_b = \alpha$, and $|C' \cap A'| \leq \alpha_a + \beta = \alpha$.

For the other direction, assume that there exists a vertex cover $C'$ of $G'$ with $|C' \cap A'|, |C' \cap B'| \leq \alpha$. We will construct a vertex cover $C$ of $G$. We begin by observing that $C'$ contains every auxiliary vertex $a_i$, for $1 \leq i \leq \beta$. Suppose otherwise, i.e., that for some fixed $i$ we have $a_i \notin C'$. Then each of the $\alpha_b + 1$ auxiliary vertices $b_{ij}$, for $1 \leq j \leq \alpha_b + 1$, is contained in $C'$. Since $\alpha = \alpha_b$, this contradicts that $|C' \cap B'| \leq \alpha$. So, we conclude that $a_i \in C'$ for each $1 \leq i \leq \beta$. We now also assume that $C' \cap B'$ consists entirely of original vertices, since the vertex set resulting from the removal of an auxiliary vertex $b_{ij}$ is still a cover, as it contains $a_i$. Now, for each $b' \in C'$, we add $b$ to $C$, and for each original $a' \in C'$, we add $a$ to $C$. Because the auxiliary vertices of $G'$ are not incident to any original edges, the original vertices of $C'$ must cover all original edges. Hence, $C$ is a vertex cover of $G$. Since $|C' \cap A'| \leq \alpha = \alpha_a + \beta$ and $C' \cap A'$ contains $\beta$ auxiliary vertices, $C' \cap A'$ must contain no more than $\alpha_a$ original vertices. Similarly, $C' \cap B'$ contains at most $\alpha = \alpha_b$ original vertices. These properties allow us to conclude that $|C \cap A| \leq \alpha_a$ and $|C \cap B| \leq \alpha_b$, as desired. $\square$

**Theorem 3.5.** COLOR-FAIR MINECC *and* COLOR-FAIR MAXECC *are* NP*-hard even in 2-regular hypergraphs with* $k = 2$.

*Proof.* Given an instance $(G = (A \uplus B, E), \alpha)$ of FAIR BIPARTITE VERTEX COVER, we construct an instance $(H = (V, E_a \uplus E_b), \tau = \alpha)$ of COLOR-FAIR MINECC with two colors ($c_A$ and $c_B$) as follows. For each edge $ab \in E$, we create a vertex $v_{ab} \in V$. We call this vertex a *conflict vertex*, and we say that it is associated with $a \in A$ and with $b \in B$. Next, for each vertex $a \in A$, we create a hyperedge $e_a$ which contains every conflict vertex associated with $a$. We color this edge $c_A$. Similarly, for each vertex $b \in B$ we create a hyperedge $e_b$ of color $c_B$ which contains every conflict vertex associated with $b$. Observe that every hyperedge is nonempty, as we may assume that $G$ has no isolated vertices. Observe also that every vertex has degree 2. We set $\tau = \alpha$.

To see that the reduction is correct, consider first a vertex cover $C$ of $G$ with $|C \cap A|, |C \cap B| \leq \alpha$. For each vertex $a \in A \setminus C$, we assign color $c_A$ to every conflict vertex associated with $a$. Similarly, for every $b \in B \setminus C$, we assign $c_B$ to every conflict vertex associated with $b$. We color remaining vertices arbitrarily. We claim that this is a valid coloring, i.e., every vertex has received exactly one color. To see this, observe that every vertex $v_{ab}$ in $V$ is a conflict vertex associated with exactly two vertices $a, b$ in $A \uplus B$. Since $C$ is a vertex cover, at least one of $a, b$ is in $C$. Hence, $v_{ab}$ is assigned exactly one color. We next claim that our assignment of colors leaves at most $\tau = \alpha$ hyperedges of any single color unsatisfied. Due to our coloring scheme, if a hyperedge $e_a$ (resp. $e_b$) associated with a vertex $a \in A$ (resp. $b \in B$) is unsatisfied, then $a$ (resp. $b$) is contained in $C$. Since $|C \cap A|, |C \cap B| \leq \alpha$, we conclude that at most $\tau = \alpha$ hyperedges of color $c_A$ (resp. $c_B$) are unsatisfied.

For the other direction, suppose that we have a coloring $\lambda\colon V \to \{c_A, c_B\}$ which leaves at most $\tau = \alpha$ hyperedges of any single color unsatisfied. We construct a vertex cover $C$ of $G$ which contains $a \in A$ (resp. $b \in B$) if and only if the hyperedge associated with $a$ (resp. $b$) is unsatisfied by $\lambda$. It is immediate that $|C \cap A|, |C \cap B| \le \alpha$. Now we claim that $C$ is a vertex cover of $G$. Consider any edge $ab \in E$. The conflict vertex $v_{ab}$ is contained in both $e_a$, which has color $c_A$, and $e_b$, which has color $c_B$. Thus, at least one of $e_a, e_b$ is unsatisfied by $\lambda$, and so $C$ contains at least one of $a, b$. Then $C$ is a vertex cover, as desired.

A simple adjustment to our construction yields the claimed hardness result for COLOR-FAIR MAXECC. Let $\beta = |A| - |B|$, and assume without loss of generality that $\beta \ge 0$. If $\beta = 0$, then there are already an equal number of edges of colors $c_A$ and $c_B$, so we make no adjustment to the construction. If $\beta \ge 3$, we add an additional $\beta$ vertices $v_1, v_2, \ldots v_\beta$, and an additional $\beta$ hyperedges $e_1 = \{v_1, v_2\}, e_2 = \{v_2, v_3\}, \ldots, e_\beta = \{v_\beta, v_1\}$, constructing a cycle with $\beta$ vertices and edges (a $C_\beta$). We assign color $c_B$ to each of these hyperedges. We now observe that in the constructed hypergraph, there are an equal number of hyperedges of colors $c_A$ and $c_B$, and every vertex has degree 2. Finally, if $\beta \in \{1, 2\}$, we add $\beta + 6$ vertices $v_1, v_2 \ldots v_{\beta+6}$. With three of these vertices we form a triangle with each edge having color $c_A$, and with the remaining $\beta + 3$ vertices we form a $C_{\beta+3}$ with each edge having color $c_B$. Once again, we observe that there are now an equal number of edges of colors $c_A$ and $c_B$ in the constructed hypergraph, and every vertex has degree 2. We set $\tau = |A| - \alpha$ if $\beta \ge 3$, or $\tau = |A| + 3 - \alpha$ otherwise. Correctness follows from a substantively identical analysis. $\square$

---

**SPARSE VERTEX COVER**

**Input:**   A graph $G = (V = V_1 \cup V_2 \cup \ldots \cup V_k, E)$.

**Task:**   Find a vertex cover $C \subseteq V$ of $G$ which minimizes $\max_{i \in [k]} \{|C \cap V_i|\}$.

---

**Theorem D.2.** COLOR-FAIR MINECC *admits a 2-approximation via reduction to* SPARSE VERTEX COVER.

*Proof.* Let $H = (V, E = E_1 \uplus E_2 \uplus \ldots \uplus E_k)$ be an edge-colored hypergraph with $k$ colors. We construct an instance $G = (V' = V'_1 \cup V'_2 \ldots V'_k, E')$ of SPARSE VERTEX COVER. For each hyperedge $e$ of color $i$, we create a vertex $v_e$ in $V'_i$. Next, for every pair of distinctly colored hyperedges $e_1, e_2$ with $e_1 \cap e_2 \ne \emptyset$ (a *bad hyperedge pair*), we create an edge $v_{e_1} v_{e_2}$. This completes the construction of $G$.

Now, let $\mathsf{opt}_G$ and $\mathsf{opt}_H$ be the optimal objective values for $G$ and $H$, respectively. We claim that $\mathsf{opt}_G \le \mathsf{opt}_H$. Suppose that $S$ is the set of hyperedges unsatisfied by some optimal coloring $\lambda$, so $\max_{i \in [k]} |S \cap E_i| = \mathsf{opt}_H$. Because $\lambda$ satisfies all edges in $E \setminus S$, $S$ must contain at least one member of every bad hyperedge pair. Hence, the set $C = \{v_e \in V' : e \in S\}$ of vertices in $G$ corresponding to hyperedges in $S$ is a vertex cover in $G$, and has the property that $\max_{i \in [k]} |C \cap V'_i| = \mathsf{opt}_H$. It follows that $\mathsf{opt}_G \le \mathsf{opt}_H$.

Finally, we show how to lift a solution. Let $C \subseteq V'$ be a vertex cover in $G$ which is 2-approximate; such a cover can be computed in polynomial time (Blum et al., 2022). Let $S = \{e \in E : v_e \in C\}$ be the hyperedges of $H$ corresponding to the vertices in $C$. Because $C$ is a vertex cover and the edges of $G$ correspond exactly to the bad hyperedge pairs of $H$, it is trivial to compute a coloring $\lambda$ which satisfies every edge in $E \setminus S$. The objective value of $\lambda$ is at worst

$$\max_{i \in [k]} |S \cap E_i| = \max_{i \in [k]} |S \cap V'_i| \le 2 \cdot \mathsf{opt}_G \le 2 \cdot \mathsf{opt}_H.$$

$\square$

**Theorem D.3.** COLOR-FAIR MINECC *admits a linear-time combinatorial $k$-approximation.*

*Proof.* We begin by proving that SPARSE VERTEX COVER admits a polynomial-time combinatorial $k$-approximation when the vertex classes $V_1, V_2, \ldots, V_k$ are disjoint and every edge has endpoints in distinct vertex classes. Let $G = (V = V_1 \uplus V_2 \uplus \ldots \uplus V_k, E)$ be an instance of SPARSE VERTEX COVER which satisfies these conditions. Let $\mathsf{opt}_G$ be the optimal objective value for SPARSE VERTEX COVER on $G$, $M \subseteq E$ be a maximal matching in $G$, and $C \subseteq V$ be $\bigcup_{e \in M} e$. We claim that $C$ is a $k$-approximate solution.

Since $M$ is maximal, $C$ is a vertex cover of $G$. Because every edge has endpoints in distinct vertex classes, for each $i \in [k]$ we have that

$$|C \cap V_i| \leq \frac{|C|}{2} \leq |M|.$$

To obtain a lower bound, we observe that because $M$ is a matching, every vertex cover of $G$ has cardinality at least $|M|$. It follows from the pigeon-hole principle that $\mathsf{opt}_G \geq |M|/k$. Thus, for each $i \in [k]$ we have that $|C \cap V_i| \leq k \cdot \mathsf{opt}_G$, as desired.

The result for COLOR-FAIR MINECC now follows from the reduction of Theorem D.2. This algorithm is the same as the vertex-cover based 2-approximation for standard MINECC previously shown by Veldt (2023), with a new analysis to prove the approximation guarantee for COLOR-FAIR MINECC. The linear runtime is obtained by applying the implicit construction process of Veldt (2023). $\qquad\square$

## E. Omitted proofs from Section 3.3

**Theorem 3.6.** *For every $\rho \in (0, \frac{1}{2}]$, there exists a polynomial-time $(\frac{1}{\rho}, \frac{1}{1-\rho})$-approximation for* PROTECTED-COLOR MINECC.

*Proof.* We begin by computing optimal fractional values $\{\gamma_e\}, \{d_v^c\}$ for LP (4).

We claim that for every pair of distinctly colored overlapping hyperedges $e, f$, $\gamma_e + \gamma_f \geq 1$. Otherwise, let $v \in e \cap f$, $c_e = \ell(e)$, and $c_f = \ell(f)$. Since $\gamma_e + \gamma_f < 1$, it follows from the second constraint of LP (4) that $d_v^{c_e} + d_v^{c_f} < 1$. Then

$$\sum_{c=1}^{k} d_v^c < 1 + k - 2 = k - 1,$$

contradicting the first constraint of LP (4).

Now we show how to color the nodes of our hypergraph $H = (V, E)$ with protected color $E_1$. For each hyperedge $e$, we set

$$\hat{\gamma}_e = \begin{cases} 1 & \text{if } e \in E_1 \text{ and } \gamma_e \geq 1 - \rho \\ 1 & \text{if } e \notin E_1 \text{ and } \gamma_e \geq \rho \\ 0 & \text{otherwise,} \end{cases}$$

and we construct the set $S = \{e \in E \colon \hat{\gamma}_e = 1\}$.

First we observe that since $\rho \in (0, \frac{1}{2}]$, $\frac{1}{1-\rho} \leq \frac{1}{\rho}$, and so

$$|S| = \sum_{e \in E} \hat{\gamma}_e \leq \frac{1}{\rho} \sum_{e \in E \setminus E_1} \gamma_e + \frac{1}{1-\rho} \sum_{e \in E_1} \gamma_e \leq \frac{1}{\rho} \sum_{e \in E} \gamma_e,$$

and

$$|S \cap E_1| \leq \frac{1}{1-\rho} \sum_{e \in E_1} \gamma_e \leq \frac{b}{1-\rho}.$$

All that remains is to compute a coloring $\lambda$ which satisfies every hyperedge in $E \setminus S$. If $S$ contains at least one member of every pair of distinctly colored overlapping hyperedges, then computing such a $\lambda$ is trivial. We conclude the proof by showing that $S$ has this characteristic. Let $e, f \in E$ with with $e \cap f \neq \emptyset$ and $\ell(e) \neq \ell(f)$. Assume toward a contradiction that $e, f \notin S$. Because $e$ and $f$ are distinctly colored, at least one is in $E \setminus E_1$. If both $e, f \notin E_1$, then

$$\gamma_e + \gamma_f < 2\rho \leq 1,$$

but we have already shown that $\gamma_e + \gamma_f \geq 1$, so exactly one of $e$ or $f$ must be in $E_1$. Assume without loss of generality that $e \in E_1$ and $f \in E \setminus E_1$. Then

$$\gamma_e + \gamma_f < 1 - \rho + \rho = 1,$$

so we have contradicted that $S \cap \{e, f\} = \emptyset$, as desired.

$\qquad\square$

**Theorem 3.7.** PROTECTED-COLOR MINECC *is FPT with respect to the total number $t$ of unsatisfied edges.*

*Proof.* We give a branching algorithm which is essentially identical to that of Theorem 3.4. For completeness, we repeat the details. Given an instance $(H = (V, E), t, b)$ of PROTECTED-COLOR MINECC, a *conflict* is a triple $(v, e_1, e_2)$ consisting of a single vertex $v$ and a pair of distinctly colored hyperedges $e_1, e_2$ which both contain $v$. If $H$ contains no conflicts, then it is possible to satisfy every edge. Otherwise, we identify a conflict in $O(r|E|)$ time by scanning the set of hyperedges incident on each node. Once a conflict $(v, e_1, e_2)$ has been found, we branch on the two possible ways to resolve this conflict: deleting $e_1$ or deleting $e_2$. Here, deleting a hyperedge has the same effect as "marking" it as unsatisfied and no longer considering it for the duration of the algorithm. We note that it is simple to check in constant time whether a possible branch violates the constraints given by $t$ or $b$; these branches can be pruned. Because each branch increases the number of unsatisfied hyperedges by 1 and WLOG $t > b$, the search tree has depth at most $t$. Thus, the algorithm runs in time $O(2^t r|E|)$. $\qquad\square$

**Theorem 3.8.** *Given a 2-regular $H = (V, E, \ell)$, two integers $b_1, b_2$, and two colors $c_1, c_2$, it is* NP-*hard to determine whether it is possible to color $V$ such that at most $b_1$ edges of color $c_1$ and at most $b_2$ edges of color $c_2$ are unsatisfied.*

*Proof.* We once again reduce from CONSTRAINED BIPARTITE VERTEX COVER, for which the input is a bipartite graph $G = (A \uplus B, E)$ along with two integers $\alpha_a, \alpha_b$, and the question is whether there exists a vertex cover $C$ with $|C \cap A| \leq \alpha_a$ and $|C \cap B| \leq \alpha_b$. We will construct the edge-colored hypergraph for which $G$ is the associated conflict graph. Formally, we construct an edge-colored hypergraph $H = (V, E_a \uplus E_b, \ell)$ with two colors $c_a$ and $c_b$ as follows. For each edge $e \in E$, we create a vertex $v_e \in V$. For each vertex $a \in A$, we create a hyperedge $e_a \in E_a$ such that $e_a = \{v_e : a \in e\}$. Similarly, for each vertex $b \in B$, we create a hyperedge $e_b \in E_b$ such that $e_b = \{v_e : b \in e\}$. For each $e_a \in E_a$ we set $\ell(e_a) = c_a$, and for each $e_b \in E_b$ we set $\ell(e_b) = c_b$. Note that $H$ is 2-regular, since each $e \in E$ has cardinality 2 and thus the corresponding vertex $v_e$ is contained in 2 hyperedges. We say that our protected colors are $E_a$ and $E_b$, and we set the associated constraints $b_1 = \alpha_a$ and $b_2 = \alpha_b$.

Thus, if $\lambda$ is a vertex coloring of $H$, $S_\lambda$ is the set of hyperedges unsatisfied by $\lambda$, and we have that $|S \cap E_a| \leq b_1 = \alpha_a$ and $|S \cap E_b| \leq b_2 = \alpha_b$, then the set $C = \{a : e_a \in C\} \cup \{b : e_b \in C\}$ has the properties that $|C \cap A| \leq \alpha_a$ and $|C \cap B| \leq \alpha_b$. Moreover, if $ab = e$ is an edge in $e$, then $e_a$ and $e_b$ are distinctly colored, with $e_a \cap e_b = \{v_e\}$. Hence, $S$ contains at least one of $e_a, e_b$, meaning that $C$ contains at least one of $a, b$, so $C$ is a vertex cover.

For the other direction, assume that $C$ is a vertex cover of $G$ with the properties that $|C \cap A| \leq \alpha_a = b_1$ and $|C \cap B| \leq \alpha_b = b_2$, so the set $S = \{e_a : a \in C\} \cup \{e_b : b \in C\}$ has the properties that $|S \cap E_a| \leq b_1$ and $|S \cap E_b| \leq b_2$. Observe that, if $e_a$ and $e_b$ are distinctly colored hyperedges which overlap, then $ab \in E$. Thus, at least one of $a, b$ is contained in $C$, and so at least one of $e_a, e_b$ is contained in $S$. Then there are no pairs of distinctly colored overlapping hyperedges in $(E_a \uplus E_b) \setminus S$. Consequently, it is trivial to compute a coloring $\lambda$ which satisfies every hyperedge not contained in $S$, meaning that it leaves at most $b_1$ hyperedges of color $c_a$ unsatisfied, and at most $b_2$ hyperedges of color $c_b$ unsatisfied.

$\qquad\square$

