# OpenReview forum: "Edge-Colored Clustering in Hypergraphs: Beyond Minimizing Unsatisfied Edges"
_ICML.cc/2025/Conference — ICML 2025 poster_

### Official Review · Reviewer_LJar · 2025-03-04

**Overall Recommendation:** 4

**Summary:**

The submission provides a range of algorithmic and complexity-theoretic contributions to the Edge-Colored Clustering problem, which has been shown to have several applications in the general area of ML.

**Claims And Evidence:**

Yes, all claims are supported by evidence (primarily proofs), although most of this is deferred to the appendix.

**Essential References Not Discussed:**

N/A.

**Experimental Designs Or Analyses:**

The experiments only have a minor, supporting role and seemed problem-free.

**Methods And Evaluation Criteria:**

The primary methods used are theoretical proofs, with some supporting implementations.

**Other Comments Or Suggestions:**

N/A.

**Other Strengths And Weaknesses:**

While the problem studied in this submission is graph-theoretical in nature, it is well-motivated and has a sufficient footprint in the machine learning community - in fact, there are several algorithmic-theory papers on the topic that appeared in recent ICML and NeurIPS conferences. In terms of technical contribution, the submission is very strong and exceeds what I would expect from an average theoretical ICML paper.

For MaxECC, the results include the first "general" approximation algorithm that is a constant-factor approximation for each hyperedge size, but also an improved polynomial-time approximation algorithm for the prominent and previously studied graph variant (i.e., the case of hyperedge size 2). I believe the latter provides great "added value" to the paper: on its own this result might not be of sufficient interest to the ML community (or even the TCS community for that matter), but it rounds up the paper's contributions for MaxECC very nicely.

In the second part of the submission, the authors turn to MinECC. While approximation algorithms for that problem were known, the authors introduce two fair/balanced variants of that problem and provide a range of algorithmic results for these (including not only novel approximation algorithms, but also parameterized algorithms and complexity-theoretic lower bounds).

In the third part of the submission, the authors support their theoretical investigation with a range of experiments. As the authors correctly state, these are only meant to have a "supporting role" and should not be seen as the main results; still, I believe they form a nice touch that complements the rest of the submission.

As can be seen from the text above, overall I view the submission's contributions very positively and would be happy to support acceptance. However, there are also some weaknesses that need to be mentioned:
1) While the motivation for MaxECC is clear, the motivation for the two new variants of MinECC is weaker - there are no references for this or similar formulations of the problem, and the variants are essentially "made up".
2) The submission attempts to convey too many results, which unfortunately spreads the story too thin and  makes the writeup rather unfocused. To be clear: I personally do not think that the inclusion of more results should be seen as a drawback on its own, but the writeup needs to be very careful to still present a coherent picture that is accessible to a reasonable fraction of the ICML community. As it stands, the paper is so dense that the introduction cannot even cover all of the individual results proved or even the individual problem variants studied. This is nicely illustrated, e.g., in the following sentence near the end of the introduction:
"Our results also include several parameterized complexity and hardness results for COLOR-FAIR MINECC and a related maximization variant where the goal is to maximize the minimum number of edges of any one color that are satisfied."

Note that the majority of the readers may not be aware of parameterized complexity theory, and so in its current state the submission might be fairly inaccessible to them (in fact, it does not even introduce parameterized complexity in the preliminaries). Later in the paper, the writeup even mentions significantly more advanced notions such as (slim) treecut-width or cutwidth... I understand that it makes no sense to define these just to have them briefly mentioned, but there should at least be a reference to point interested readers to some literature.

Overall, I have a very high opinion of the paper's results and technical contributions, but given these weaknesses I personally see this more as a solid "accept" than a "strong accept".

## Update after rebuttal
Thank you for the responses; I maintain my score.

**Questions For Authors:**

You introduce two novel MinECC variants that take into account fairness/balance aspects (Lp-Norm MinECC and Protected-Color MinECC) and provide some nice results for both of these. But on a higher level, I wonder why it makes sense to have the fairness part focus on MinECC even though the first part (which is better linked to previous work) focuses exclusively on MaxECC. Maybe the authors could comment on this discrepancy - is it because there's no nice "natural" formulation of fair MaxECC, or is it because getting results for such a formulation are more elusive?

**Relation To Broader Scientific Literature:**

The key contributions are related to broader scientific literature.

**Theoretical Claims:**

I did not check the details of all proofs, but those I did see seemed reasonably well-written and problem-free.

---

> ### Author Rebuttal · Authors · 2025-03-31
>
> Thanks for the helpful feedback on our manuscript. With regards to the motivation please see our response to the reviewer Tzam for details on our motivation for these variants.
>
> Furthermore, we agree that some sections are quite dense on results and sparse on details. We wished to provide more details and a less terse write-up (in particular, a more comprehensive introduction), but due to the page limit we ended up having to push many results to the appendix. We decided it is better to include all results, since there are readers who care about them (e.g., the parameterized complexity results). If the paper is accepted, we plan to use the extra 1 page of space to address these exact concerns you are raising (which we agree with). In particular, an extra page of space will allow us to provide a fuller introduction and overview of our results, provide needed background on parameterized complexity for a broad audience, and fill in some other details that are dense in the current manuscript.
>
> Finally to answer the question you posed, this is answered for Lp-norm ECC by our hardness result for the special case of Color-Fair MaxECC (see Section 3.2). One could argue that Color-Fair MaxECC is in fact more natural than Color-Fair MinECC in some settings (to answer part of your question). However, Theorem 3.3 implies that it is NP-hard to approximate this objective to within any factor even in graphs (in fact, just paths). Hence, as you wondered, “results for such a formulation are more elusive”. Regarding Protected Color ECC, we agree that a maximization variant could also be interesting and our hardness results do not rule out approximations for bounded hyperedge sizes. This is an interesting direction for future work, but as of yet we have not established any results.

---

### Official Review · Reviewer_qaps · 2025-03-12

**Overall Recommendation:** 3

**Summary:**

The authors study the edge-colored clustering (ECC) problem in hypergraphs. They generalize MaxECC from graphs to hypergraphs, and present an approximation algorithm of factor $(2/e)^r (r+1)^{-1}$, where $r$ is the maximum number of nodes that are allowed in a single hyperedge. A slight modification of this algorithm provides a factor of 154/404 for graph MaxECC. They also study three variants of ECC, that are, $\ell_p$-norm ECC, color-fair ECC and protected-color ECC, and give a series of algorithmic and hardness results on them. Experimental results demonstrate that their algorithms have much better results in practice than the theoretical worst-case guarantees.

**Claims And Evidence:**

Yes, I think so.

**Essential References Not Discussed:**

Yes, several key related works have been cited properly in this paper.

**Ethical Review Concerns:**

No ethical review concern.

**Experimental Designs Or Analyses:**

No baseline method is introduced in the experiments. But I’m not sure whether there is.

**Methods And Evaluation Criteria:**

Yes, except the benchmark datasets for which there is not too much information provided in the paper, e.g., what the nodes and hyperedges represent, what the colors mean, etc.

**Other Comments Or Suggestions:**

1. It’s better to introduce more about the benchmark hypergraphs from the references (Veldt, 2023; Amburg et al., 2020), especially when they are used for the variants of the ECC, as such datasets are not very common after all. The information in Table 2 is insufficient.
2. Line 411, Column 1, “Table 4” should be “Table 1”.

**Other Strengths And Weaknesses:**

S1. The theoretical analyses are solid, and the conclusions are rich.

W1. I am not sure how many new technical contributions are developed, especially compared with (Veldt 2023). LP seems to be the common starting point for ECC algorithms in each prior work. Randomized rounding and global color ordering techniques have also been used. I understand that the two problems MaxECC and MinECC are different from the perspective of approximation algorithm, but I am not clear what the essential difference is when the same techniques are used for both of them.

W2. I am not clear for the motivation and practical significance of the variant problems, or they are just in a mathematical sense.

**Questions For Authors:**

1. Can you give a comparison with the techniques that are used in (Veldt, 2023)? See W1 for more details on this question.
2. Is there any practical example to demonstrate the significance of the variant problems of ECC?
3. Can you explain more about the benchmark datasets and your manipulation on them for your experiments?

**Relation To Broader Scientific Literature:**

This research fits well in the study line of ECC on (hyper)graphs. It has the first approximation algorithmic result of MaxECC for the hypergraph case and improves that (even slightly) for graph case. It also complements some algorithmic and hardness results on three variants of MinECC for hypergraphs.

**Theoretical Claims:**

I have only checked the proof of Theorem 2.1 for the basic algorithm, and had a quick review for the variants. Every algorithmic result is starting from some kind of LP description of the relative problem, while LP-rounding and randomization are main techniques in analysis. All conclusions are easy to understand.

---

> ### Author Rebuttal · Authors · 2025-03-31
>
> Thanks for your detailed review and feedback on our manuscript.
>
> Regarding the “significance of the variant problems for ECC”, please see our response to reviewer Tzam, where we provide more details on motivation we could have been clearer about in our submission.
>
> Thanks for your question about comparing our techniques to those of Veldt 2023. There are indeed strong high-level similarities (LP relaxation + global color ordering + randomized rounding), but the difference between MaxECC and MinECC is significant enough that the algorithm of Veldt 2023 can be shown to completely fail for MaxECC. Furthermore, the way to make an algorithm work for MaxECC requires quite a bit of new technical machinery, and as a result the overall techniques and proof for our algorithm substantially differ from the technical contribution Veldt 2023.
>
> To explain this in more depth, note the following important differences between the randomized LP rounding MinECC algorithm of Veldt 2023 and our randomized LP rounding algorithm for MaxECC:
>
> (1) We have a threshold (determining how colors want nodes) for each color, while Veldt 2023’s algorithm uses a single threshold that applies to all colors at once.
>
> (2) Especially for the graph case, we have to distinguish between whether a color is “strong” or “weak” for a node
>
> (3) We draw each color threshold uniformly from [0,1]. Veldt 2023 draws the threshold within a bounded subset in the interval [0,1]: either [⅓, ½] or [1/4 , 1/2] or [⅛, ½] depending on the relationships between k and r. Note that these intervals appear at face value to be different from what is written in Veldt 2023, but that is only because Veldt was considering flipped LP variables (essentially, for our LP variables $x_{u}^c$ Veldt was working with $1-x_{u}^c$ )
>
> Focusing on difference (3), choosing a threshold that is bounded away from 0 means that Veldt could essentially ignore hyperedges where the $z_e$ variable is very small, even if $z_e > 0$. For the MinECC formulation, this means there is a variable $x_e = 1- z_e$ that is close to 1, and note that the LP relaxation for MinECC has objective min
> $\sum_{e} x_e$. When $x_e$ is large, then we can just delete (leave unsatisfied) that edge with probability 1, and the “mistake” (for the MinECC objective) is paid for in terms of the (large) value of $x_e$ in the LP. Thus, Veldt 2023 can focus on edges e where $z_e$ is not too small (i.e., $x_e$ is not too large). This is essential for the analysis because this implies useful bounds on the number of different colors that can want each node in the hyperedge, which in turn leads to simpler bounds on the probability of making the edge unsatisfied.
>
> However, for MaxECC, if we choose a threshold that is bounded away from zero when deciding how nodes want colors, this means that if an edge e has an LP variable $z_e$ that is positive but close to zero, the algorithm of Veldt 2023 is guaranteed to delete it, meaning that the probability that the edge $e$ is satisfied is 0. This completely ruins the strategy of proving that for every edge e, the probability of satisfying e is at least $p z_e$ for some approximation value $p$.
>
> Because of this, a different algorithm (with a very different proof) is necessary. First of all, we must consider thresholds that can be arbitrarily small (hence we draw them from [0,1]). This then makes the analysis much more complicated, because now we do not have any convenient bounds on the number of colors that want nodes in an edge (recall that this was crucial for the analysis of Veldt 2023). This is what leads to other needed differences in the algorithms (in particular, differences (1) and (2) listed above). In terms of the proofs themselves, our (often quite complicated) supporting lemmas (Lemmas A.2-A.6) and the ways they fit together to provide full proofs, deviate significantly from the technical details of Veldt 2023 for MinECC. (Note also that Observations 1 and 3 are simple results that are analogous to observations used by Veldt 2023, but Observations 2 and 4 are unique to our algorithm and analysis for MaxECC).
>
> In regards to the datasets, we used a suite of datasets that have been described in detail in previous work on ECC (Amburg et al WWW 2020). Because these datasets have been described in detail previously and because of space constraints, we omitted a more detailed description. However, we understand that more details would be nice so that readers do not have to refer to other work, and we are happy to include more details in the appendix. Additionally, if the paper is accepted, that would provide us with 1 page of additional space we may be able to use. Some of this 1 page of space is also needed to expound on parameterized complexity (see response to reviewer LJar), but we may also be able use some of this space to briefly expound on the datasets even in the main text.
>
> Thanks finally for noticing the typo on line 411.

---

### Official Review · Reviewer_Hv7K · 2025-03-13

**Overall Recommendation:** 4

**Summary:**

This paper studies variants of the Edge Colored Clustering problem. The input consists of a hypergraph where the hyperedges are equipped with a color each. The goal is to find a coloring of the vertices. A hyperedge e is satisfied if all its vertices have the same color as e; otherwise e is unsatisfied.
An important variant is MaxECC where the goal is to maximize the number of satisfied edges. The paper provides the first approximation algorithm hor hypergraph MaxECC. Morover, an improved approximation factor for the graph variant is provided.
Moreover, also balanced and fair variants of ECC are studied and mainly approximation algorithms are provided.
Finally, a proof-of-concept implementation is provided showing the effectiveness of the newly proposed algorithms.

**Claims And Evidence:**

For me, all claims are convincing. Also the problem and its variants have a good motivation.

**Essential References Not Discussed:**

I think all relevant papers are discussed.

**Experimental Designs Or Analyses:**

The experiments are only a proof-of-concept implementation. But these experiments are sufficient to outline the effectiveness of the new approximation algorithms.

**Methods And Evaluation Criteria:**

Yes.

**Other Comments Or Suggestions:**

line 99 column 2: I think w should be part of the definition of H

line 130 column 1: please provide some references

line 190 column 2: Can you add a short description why the proof does not work for r>2?

line 236 column 1: I got a good intuition for the used prioritization, but I didn't get an intuition for ``strong''. Please add a few lines there for more intuition.

line 311 column 2: I think many more parameters are open; and an interesting one might be the max leaf number.

Theorem 3.9 If I understand it correctly, I think this theorem should also hold for 2-regular graphs? If yes, please mention this in the theorem.

line 411 column 1: Table 4 >> Table 1

line 527 that that >> that the

line 1134: the forward reference to section 3.2 is not very nice; please define it here too.

line 1472: I think Protected-Color MinECC should be NP-hard for b=0. I think this should be mentioned somewhere.

table 2: explain shortly the meanings of n,m,r,k

**Other Strengths And Weaknesses:**

other strength:
I liked that ideas from the hypergraph algorithm are adapted to the graph case and this then yields a better approximation factor than what was previously known.

other weaknesses:
The FPT-algorithms are very simple. It would be interesting to have matching lower bounds, eg no 2^o(t) algorithms are possible.
But I think this is not a real weakness. I see these FPT results as a small bonus and that the main results are the approximation algorithms.

**Questions For Authors:**

Q1: I don't understand the second equation in Observation 3. I think you need to multiply the probabilities? I think it is used like this in line 196 column 1.

Q2: Where is the proof of Theorem 3.2

Q3: Can't you simplify the proof of Theorem 3.3 significantly, if you additionally assume that each clause has exactly 3 literals?

Q4: Are you aware of any ETH lower bounds for your FPT-algorithms with respect to t?

Q5: Are there any inapproximation results for the balanced and fair variants of your problem?

**Relation To Broader Scientific Literature:**

In the introduction a very good overview of related papers and previously known approximation factors is provided.

**Theoretical Claims:**

I did not check all proofs in detail (especially the long proofs about the approximation guarantee), but I think I got a rough understanding of the ideas.

---

> ### Author Rebuttal · Authors · 2025-03-31
>
> Thanks for the detailed review! We agree the FPT algorithms are simple, and intended only as small supporting results. Also, please see our answer below regarding ETH bounds.
> Thank you also for your line-based comments and suggestions. We will revise the manuscript accordingly. We’ll focus the majority of our space on your 5 specific questions.
>
> Q1: This observation considers the probability that every node in an edge wants the edge’s color. A node wants a color if the LP variable for that node-color pair is greater than the random threshold for that color. What we are observing is that a single random threshold is used for the entire color, so in order for all nodes in an edge to want a color, it is enough for the node corresponding to the smallest LP variable to want the color. Note that the random variables in question are not independent, so we do not want to multiply the probabilities
>
> Q2: We omitted this proof, since the text preceding the theorem statement contains the main idea. However, we would be happy to add a formal proof to the appendix. All that is missing is a standard rigorous treatment of the limit analysis.
>
> Q3: This is good intuition, but reducing from 3SAT (with our construction, at least) only proves that Color-Fair MinECC is NP-hard when the rank of the hypergraph is at least 3, i.e., we would leave the graph case open. To understand why, observe that in our reduction we assume that each literal appears no more than twice. This assumption is crucial in bounding the size of constructed edges by 2 (each variable-clause pair has no more than 2 associated conflict vertices). If we instead reduce from 3SAT, then by the results of Tovey (Discrete Applied Math; 1984), we must assume that some variable appears at least 4 times, meaning that a literal may appear thrice. One might wonder whether the reduction of Tovey can be adapted to guarantee that literals appear at most twice. This may be possible, but it is not straightforward (we tried).
>
> Q4: For Color-Fair MinECC, observe that the reduction of Theorem 3.3 creates an instance with O(m) total edges, where m is the number of clauses in the SAT instance. Clearly, the parameter t is thus also in O(m). So, conditioned on an ETH-bound for the particular variant of SAT from which we reduce, we can obtain one for Color-Fair MinECC.
>
> Regarding the relevant SAT variant, at a glance it appears that the ETH excludes any 2^o(m) algorithm. To formalize this, one needs to make a few relatively simple arguments about the reduction of Tovey (Discrete Applied Math 1984). Having not worked out all of the details formally, for now we conjecture strongly that, under the ETH, Color-Fair MinECC does not admit any 2^o(t) algorithm.
>
> For Protected-Color ECC, first observe that the problem generalizes standard ECC. We believe that no ETH bound for ECC has ever appeared in the literature. However, it is clear that one can be obtained. For example, the reduction from Vertex Cover given by Veldt 2023 excludes any 2^o(t) algorithm for ECC (each hyperedge corresponds to a unique vertex in the Vertex Cover instance). The same bound then applies to Protected-Color ECC.
>
> The aforementioned Vertex Cover reduction creates an ECC instance with unbounded rank. It is thus open (and perhaps more interesting) to seek an ETH bound for ECC when the rank is bounded, most prominently in the graph (rank 2) case. Perhaps an ETH bound can be inferred from one of many reductions given by Kellerhals et al. (AAAI '23).
>
> Q5: For Color-Fair MinECC we are not aware of any approximation bound, though we agree that this is interesting. In particular, it is interesting to ask whether Color-Fair MinECC can be approximated as well as (or even better than) standard MinECC, for which Veldt 2023 gave a UGC-hardness bound.
>
> Further progress beyond our given 2-approximation is likely to require a new perspective. To see why, observe that we obtain 2-approximations for Color-Fair MinECC via two distinct strategies. First, we round the LP given at the beginning of Section 3.1 (appropriately rewritten for the mini-max p → infinity case). We cannot tighten this technique, since we already match the integrality gap. Second, we obtain a 2-approximation via reduction to Sparse Vertex Cover (see line 333 column 1, and Thm C.2). Improving the bound yielded by this reduction would require a better-than-2-approximation for Sparse Vertex Cover, which would refute the UGC. We mention these points only to emphasize that, while we do not have approximation lower bounds for Color-Fair MinECC, our results are tight with respect to the techniques used.
>
> For Protected-Color ECC, we can apply the UGC-conditioned lower bound for standard ECC from Veldt 2023. However, it is interesting and open to determine whether Protected-Color MinECC can be approximated as well as standard MinECC. Finally, we mention (line 298, column 2) that unless P = NP, no poly-time multiplicative approximation for Color-Fair MaxECC exists.

---

> > ### Comment · Reviewer_Hv7K · 2025-04-03
> >
> > Thank you for your answers!
> >
> > about Q3 (and partiallyQ4):
> >
> > I think the problem (3,B2)-SAT is want you want: The 3-SAT-variant where each literal positive/negative appears exactly twice.
> > This problem is NP-hard, see the paper 'Approximation hardness of short symmetric instances of MAX-3SAT' (Theorem 1).
> > Also, I think for this problem a no $2^{o(n+m)} lower bound under the ETH is known (or should follow easily by considering their reduction).

---

> > > ### Author Response · Authors · 2025-04-08
> > >
> > > Thanks for the follow-up comment and the helpful reference! This sounds promising; we’ll take a close look. At a glance, it appears to us that you are correct about the ETH bound following from their reduction.
> > >
> > > Thanks once again for the detailed review and helpful suggestions for improving the work.

---

### Official Review · Reviewer_Tzam · 2025-03-14

**Overall Recommendation:** 2

**Summary:**

This submission studies a clustering problem where one wants to cluster vertices of a hypergraph by coloring them while approximately maximizing the number of hyperedges all whose vertices get a hyperedge-specific color (those hyperedges are called satisfied). It gives the first algorithm for this problem on general hypergraphs the approximation ratio of which gets worse with increasing maximum hyperedge size but actually beats the previous best ratio for hyperedge size 2. The algorithm, like older ones for the graph setting, is based on randomized rounding of an LP solution. As is often the case, the idea behind this rounding is natural but analyzing its quality involves some carefully combined ideas.
Additionally, the submission proposes variants of the dual problem for minimizing the number of unsatisfied hyperedges that put hard constraints on the number of allowed unsatisfied hyperedges of a specific protected color or require that the number of unsatisfied edges of each color be balanced and devises some basic hardness results and algorithms for them. The latter are partially implemented and tested.

**Claims And Evidence:**

The only content I am suspicious of is the motivation behind defining the fair variants of the problem. The described potential applications sound more like the size of the clusters should be lower bounded rather than the number of unsatisfied clusters balanced. If one were to reformulate that application as wanting each type of task to be not be doable because of a certain assignment of workers to roles one would also consider it more sensible to somehow view this relative to the number of tasks of a type there are in total. In general, I would need some more convincing arguments why the suggested fair problems are interesting, even from a theoretical standpoint.

**Essential References Not Discussed:**

I am not aware of any.

**Experimental Designs Or Analyses:**

I did not check them beyond reading their description and thinking about whether they make sense on a superficial level but these are also not the core of the contributions.

**Methods And Evaluation Criteria:**

The methods seem appropriate.

**Other Comments Or Suggestions:**

-L026: best known (no hyphen)
-L058: Maybe say here what this gap is
-L135: Maybe add r is an arbitrary constant
-L138: line of previous research
-At the beginning of section 2.1.: at some point you should fix z_e and such to be values of the variables in an arbitrary fixed optimal solution of the LP (otherwise z_e is not defined in the statements of the observations and such).
-L166: Obtain optimal values for {z_e,x_v^c} in the LP relaxation.

**Other Strengths And Weaknesses:**

In terms of topic, this could be in scope for ICML but the problem seems a bit niche for the broad audience (this impression seems to also be confirmed by looking at where most closely related work was published). Apart from that, the results and included proofs are also generally well-presented.

Because I am not convinced of how relevant the general problem and even more so the introduced variants are, I am scoring this as a weak reject but am happy to discuss this assessment.

**Questions For Authors:**

Can you argue a bit more why the introduced fair variants of the problem are of interest?

Similarly, your later results seem a bit arbitrary. Can you explain why you chose to focus on those, e.g. some approximation, some approximation results and some hardness results in which you do not mention hardness of approximation? Did you try to get a comprehensive view in any of these directions?

**Relation To Broader Scientific Literature:**

This falls into the research direction on edge-colored clustering, improves a line of results there and tries to open new research directions about the problem.

**Theoretical Claims:**

I read everything except the appendix.

---

> ### Author Rebuttal · Authors · 2025-03-31
>
> Thanks for your feedback and questions!
>
> **Motivation for ECC variants.** We highlight 2 points we could have done better addressing in the main text.
>
> (1) Correlation Clustering variants
>
> One key motivation for our fair/protected ECC variants is that these are directly analogous to questions that have been extensively studied for variants of the closely related correlation clustering (CC) problem. Puleo and Milenkovic (ICML 2016) considered a mini-max objective (minimize the maximum number of mistakes incident to any one node) and a more general Lp-norm variant of correlation clustering. Subsequent papers on these CC variants include:
>
> Charikar et al. (IPCO '17)
>
> Kalhan et al. (Neurips '19)
>
> Jafarov et al. (ICML '21)
>
> Davies et al. (ICML '23)
>
> Heidrich et al. (AISTATS '24)
>
> Davies et al. (ICALP '24)
>
> Unlike the rich literature on correlation clustering, prior work on ECC has focused only on objectives that consider the total count of (un)satisfied edges. Our motivation is to understand whether algorithms for this simplest variant work only in this setting, or if we can identify algorithmic strategies that work (or fail) for alternative objectives.
>
> (2) Resource allocation, team formation
>
> We are also interested in potential applications to resource allocation (motivation from the first graph ECC paper, Angel et al 2014) and team formation problems that relate to the work of Amburg et al 2022, but go beyond the explanation we provided in our introduction. For team formation, one can imagine nodes representing workers, and colors representing assignments to work teams (possibly encoding different physical work sites). An edge of a certain color is then a task that can be completed if and only if a specific set of individuals are assigned to that same group or site. In this case, a natural goal is to partition workers to balance the number of tasks that can be accomplished at different work sites. Here it’s not enough to just balance the number of workers in each group—one must consider the collection of tasks that can be completed by workers assigned to a team.
> (This motivation aligns more with Color-Fair MaxECC. However, the latter is NP-hard to even approximate. Color-Fair MinECC provides roughly the same goal if the number of edges of each type is balanced, and we show it is amenable to approximations.)
>
> We stress that this direction of exploration is still in initial phases, which is why we did not focus on it extensively in our submission. The present paper is more directly motivated to establish better theoretical foundations of ECC algorithms for a broader class of objective functions, analogous to the rich literature on correlation clustering.
>
> **Regarding results for fair/balanced ECC.** Our goal was to get as complete an understanding as possible of (i) the approximability and (ii) parameterized complexity of our fair/balanced objectives, especially for Color-Fair ECC. We agree that there are still open questions that could be explored, but we sought to provide as full of an answer as possible.
>
> Regarding FPT results, we chose to explore the same collection of parameters considered by Kellerhals et al. (AAAI 2023), who provided a comprehensive study of the parameterized complexity for standard ECC. This uncovered fundamental differences between FPT results for standard ECC and Color-Fair ECC (see text after Theorem 3.3).
>
> Note that we did include some approximation hardness results, in that Color-Fair MaxECC is NP-hard to approximate to within any factor (see line after Theorem 3.3). We agree that approximation hardness results for Color-Fair MinECC and Protected ECC would also be interesting, but ultimately we were not able to settle all open questions in one paper. These are interesting directions for future work.
> See also our response to Question 5 of reviewer Hv7k regarding approximation hardness and ways in which our results are tight.
>
> We agree that there is a lot going on in the second half of the paper in various directions. We could have chosen to discard some of these results, but ultimately we felt it would be better to provide as many answers to natural questions as we could.
>
> **Related work.** The reviewer argued that “the problem seems a bit niche for the broad audience (this impression seems to also be confirmed by looking at where most closely related work was published).” However, we would like to point out that the most related papers are published at ICML or similar venues. This includes hypergraph ECC papers:
>
> Veldt (ICML '23)
>
> Amburg et al (WWW '20)
>
> Amburg et al (SIAM Data Mining '22)
>
> Kellerhals et al (AAAI '23)
>
> Crane et al (WSDM '24)
>
>
> and papers on approximation algorithms for the closely related problem of correlation clustering in edge-colored graphs: Bonchi et al (KDD 2012), Anava et al (WWW 2015), Klodt et al (KDD 2021), Xiu et al (Neurips 2022)
>
> Thanks finally for the additional Comments and Suggestions. We will address these in an updated version of the paper.

---

> > ### Comment · Reviewer_Tzam · 2025-04-05
> >
> > Many thanks for your response!
> > The motivation for the ECC variants is now clearer to me and I would suggest incorporating in particular (1) into the manuscript; at least I as a reader would have appreciated that.
> >
> > If you tried and were not able to get some more comprehensive results, I think it is always nice to indicate some key difficulties which follow-up work needs to resolve. Also, as reviewer Hv7K points out, some hardness results seem quite easy to obtain. I still think that whole part of the paper could benefit from a more focused structuring and less of being presented like a arbitrary accumulation of results.
> >
> > As for the most closely related work, I would view that as the series of improvements in the graph case, since you are doing something similar in the hypergraph setting. Those publications appeared at less broad venues.

---

> > > ### Author Response · Authors · 2025-04-08
> > >
> > > Thanks for your reply!
> > >
> > > *"The motivation for the ECC variants is now clearer to me and I would suggest incorporating in particular (1) into the manuscript; at least I as a reader would have appreciated that."*
> > >
> > > Yes, we will definitely be incorporating the motivation from (1) into the manuscript; we agree that in our original submission we did not do a good enough job highlighting this relevant previous work and motivation.
> > >
> > > *"If you tried and were not able to get some more comprehensive results, I think it is always nice to indicate some key difficulties which follow-up work needs to resolve. Also, as reviewer Hv7K points out, some hardness results seem quite easy to obtain. I still think that whole part of the paper could benefit from a more focused structuring and less of being presented like a arbitrary accumulation of results."*
> > >
> > > We agree that the second half of the paper will benefit from a focused restructuring. If the manuscript is accepted, this will give us one extra page of space, and we have already identified this as the section that could most use it. In particular, we’ll do a better job highlighting why our parameterized complexity results do already provide pretty comprehensive answers regarding how the parameterized complexity of Color-Fair ECC relates to standard ECC (and yes, the hardness results highlighted in our discussion with reviewer Hv7K can help strengthen this as well!). We’ll also do a better job highlighting the challenges that would need to be overcome in order to get better approximations for our problems.
> > >
> > > We’ll also be relegating some of the minor side results to the appendix to help better focus this section (e.g., the linear-time k approximation for Color-Fair ECC).
> > >
> > > *“As for the most closely related work, I would view that as the series of improvements in the graph case, since you are doing something similar in the hypergraph setting. Those publications appeared at less broad venues.”*
> > >
> > > Understood, thanks for the clarification. Those publications on graph ECC are indeed at less broad venues, but given the many papers on algorithms for ECC at ICML and similar venues, we believe this is the right venue for our work. We hope that the relevant related work on correlation clustering that we highlighted in our response also helps address this concern.
> > >
> > > Once again, thanks for your detailed review and for taking the time to provide some additional follow-up comments.

---

### Decision · Program_Chairs · 2025-05-01

**Decision:**

Accept (poster)

**Comment:**

This paper makes interesting progress on a clustering-like problem. Here is one basic problem that receives much attention in the paper. We are given a hypergraph H = (V, E), i.e., a set family where each member of E (i.e., each “hyperedge” or “edge”) is a subset of the set of “vertices” V; we also have a non-negative weight for each e in E (note again that e is a hyperedge). Each hyperedge is also given a “color” c from the range {1, 2, …, k}. The goal is to assign each vertex v in V a color from the same range {1, 2, …, k}, so that the weighted sum of “satisfied” hyperedges---those e in E for which each of e’s elements gets the same color as e---is maximized. Parametrizing by the “dimension”---maximum size of a hyperedge---r of H, the paper develops good approximation algorithms with approximation ratio a function of r for this problem by a novel randomized rounding scheme to the natural linear-programming relaxation of the problem. Various improvements and additional results are also developed: e.g., a related minimization problem, the useful special case r = 2 (where H is a graph), and “fair” versions.

I know several rounding theorems and their subtleties and was surprised positively by the nice rounding scheme in the paper. It will likely have further consequences.

The paper is in the area of “fair combinatorial optimization” and although connected to ML through clustering, the fit with ICML is not strong, and nor is the fit absent.